# Proteomics reveal biomarkers for diagnosis, disease activity and long-term disability outcomes in multiple sclerosis

Julia Åkesson [1,2,8], Sara Hojjati[3,8], Sandra Hellberg[1,3], Johanna Raffetseder [3], Mohsen Khademi [4], Robert Rynkowski[5], Ingrid Kockum [4], Claudio Altafini [6], Zelmina Lubovac-Pilav [2], Johan Mellergård [5], Maria C. Jenmalm [3], Fredrik Piehl [4], Tomas Olsson[4,8], Jan Ernerudh [7,8] & Mika Gustafsson [1,8] ✉

Sensitive and reliable protein biomarkers are needed to predict disease trajectory and personalize treatment strategies for multiple sclerosis (MS). Here, we use the highly sensitive proximity-extension assay combined with next-generation sequencing (Olink Explore) to quantify 1463 proteins in cerebrospinal fluid (CSF) and plasma from 143 people with early-stage MS and 43 healthy controls. With longitudinally followed discovery and replication cohorts, we identify CSF proteins that consistently predicted both short- and long-term disease progression. Lower levels of neurofilament light chain (NfL) in CSF is superior in predicting the absence of disease activity two years after sampling (replication AUC = 0.77) compared to all other tested proteins. Importantly, we also identify a combination of 11 CSF proteins (CXCL13, LTA, FCN2, ICAM3, LY9, SLAMF7, TYMP, CHI3L1, FYB1, TNFRSF1B and NfL) that predict the severity of disability worsening according to the normalized age-related MS severity score (replication AUC = 0.90). The identification of these proteins may help elucidate pathogenetic processes and might aid decisions on treatment strategies for persons with MS.

Achieving personalized multiple sclerosis (MS) treatment strategies requires more refined data than evaluation of relapse rate, disease progression, and measurements of magnetic resonance imaging (MRI) activity in early disease stages[1]. The comprehensive investigation of MS biomarkers, including their validation on a completely new cohort, remains exceptionally rare. A recent meta-analysis study has shown that less than 8% of all studies have adopted this stringent methodology in order to establish the robustness and generalizability for

modeling MS[2]. To identify new MS biomarkers, extensive discovery approaches are required, such as large-scale proteomics[3] which has shown significant potential in investigating cerebrospinal fluid (CSF) to elucidate various aspects of the disease[4]. The proximity extension assay (PEA), recently combined with next-generation sequencing (PEA-NGS or Olink Explore), allows for large-scale investigation of almost 1500 proteins in a small volume with high sensitivity and accuracy[5–7]. This technology has provided opportunities for identifying protein

[1]Bioinformatics, Department of Physics, Chemistry and Biology, Linköping University, 581 83 Linköping, Sweden. [2]Systems Biology Research Centre, School of Bioscience, University of Skövde, 541 28 Skövde, Sweden. [3]Division of Inflammation and Infection, Department of Biomedical and Clinical Sciences, Linköping University, 581 83 Linköping, Sweden. [4]Neuroimmunology Unit, Department of Clinical Neuroscience, Center for Molecular Medicine, Karolinska University Hospital, Karolinska Institute, 171 76 Stockholm, Sweden. [5]Department of Neurology, and Department of Biomedical and Clinical Sciences, Linköping University, 581 83 Linköping, Sweden. [6]Division of Automatic Control, Department of Electrical Engineering, Linköping University, 581 83 Linköping, Sweden. [7]Department of Clinical Immunology and Transfusion Medicine, and Department of Biomedical and Clinical Sciences, Linköping University, 581 83 Linköping, Sweden. [8]These authors contributed equally: Julia Åkesson, Sara Hojjati, Tomas Olsson, Jan Ernerudh, Mika Gustafsson. ✉e-mail: mika.gustafsson@liu.se

biomarkers[5,8–10] that are otherwise difficult to detect due to their low abundance in body fluids.

MS is a chronic inflammatory and degenerative disease of the central nervous system (CNS), causing inflammation, demyelination and neuroaxonal damage[11]. Early initiation of treatment, particularly with high-efficacy therapies, has been associated with better clinical outcomes and can delay neurological disability progression[12–15]. On the other hand, unnecessary treatment must be avoided[16]. Since early treatment affects long-term disability outcome, it is likely that disease-associated pathways leading to demyelination and neuroaxonal damage are present already at early stages of the disease. This, in turn, would allow for the discovery of early biomarkers able to predict subsequent disease progression and to provide optimal treatment strategies for each person.

Immunological and neurological disease processes can impact the composition of circulating body fluids[9]. As a result, changes in protein levels in blood and CSF can be used as biomarkers for disease recognition and disease activity[10,17–19]. Most protein biomarkers of relevance in MS have been identified in CSF[20], while only a few candidates have been identified in plasma[10]. Since blood samples are much easier to collect and can be collected repeatedly as compared to CSF, plasma makes a more attractive option for biomarker discovery. However, potential biomarker proteins are generally less abundant in plasma than in CSF[21]. Furthermore, it remains unclear how well protein levels in plasma reflect disease-relevant processes taking place in the CNS, and in general, plasma and CSF protein levels do not correlate[22].

In this study (see overview in Fig. 1), we use the highly sensitive and specific PEA-NGS technology to measure the expression of 1463 proteins in paired CSF and plasma samples from two well-defined cohorts of persons with MS (pwMS) in the early stages and healthy controls (HC). We identify a set of differentially expressed MS-relevant proteins and test their ability to predict, either individually or in combination, short-term disease activity and long-term confirmed disability worsening.

## Results

### Proteins in CSF were differentially expressed in MS versus HC in two independent cohorts

We analyzed protein expression levels of 1463 proteins in both CSF and plasma samples from 143 pwMS in early stages of the disease and 43 HC. The pwMS were divided into a discovery cohort (92 pwMS and 23 HC from Linköping University Hospital) and a replication cohort (51 pwMS and 20 HC from Karolinska University Hospital; Table 1). Plasma samples from 21 pwMS in the replication cohort had higher expression of several protein markers known to be affected by sampling and handling variability[23] and were therefore excluded from further analysis (see Supplementary Fig. 1 and Supplementary Fig. 2). Using linear model t-test (Limma analysis) we first tested if proteins were differentially expressed between pwMS in a relapse or not, on treatment or not within 3 months before baseline sampling, or based on disease duration at baseline sampling. No differentially expressed proteins (DEPs) between these groups were found (false discovery rate (FDR) < 0.05; see "Methods"). Therefore, all pwMS were included in the following analyses.

Next, we compared the protein expression in CSF between all pwMS and the HC and found a clear separation by principal

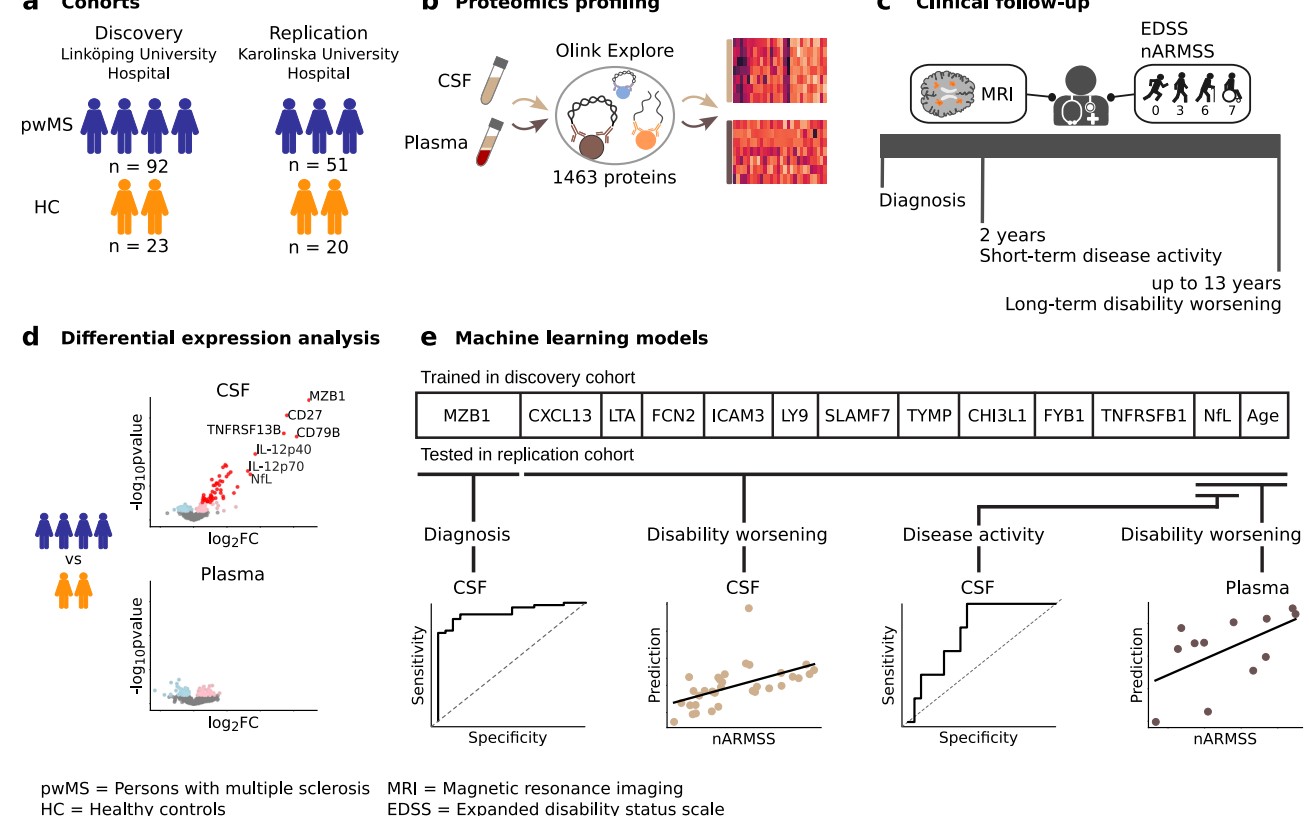

**Fig. 1 | Overview of the study. a** Prospective longitudinal study of two Swedish cohorts of persons with MS (pwMS) in the early stages and healthy controls (HC). **b** Proteomics profiling of cerebrospinal fluid (CSF) and plasma samples of all pwMS and HC at baseline. **c** Clinical examination of pwMS during a follow-up of up to 13 years. **d** Differential expression analysis, performed with a two-sided linear model t-test (Limma analysis), to find MS biomarker candidates. **e** Building machine learning models for identification of protein MS biomarkers for diagnosis (logistic regression model), prediction of short-term disease activity (logistic regression model), and prediction of long-term disability worsening (linear regression model).

**Table 1 | Baseline characteristics of persons with MS (pwMS) and healthy controls (HC)**

| | | Discovery cohort | | Replication cohort | | p-value* Discovery vs. Replication |
|---|---|---|---|---|---|---|
| | | MS | HC | MS | HC | |
| Cohort size | n | 92 | 23 | 51 | 20 | NA |
| Sex** | F/M | 67/25 | 18/5 | 39/12 | 10/10 | 0.69 |
| Age** (years) | Median (range) | 31 (16–64) | 32 (22–64) | 32 (18–54) | 30 (22–47) | 0.80 |
| CSF data** | | | | | | |
| CSF cell count | Median (range) | 4.7 (0–125) | 2.1 (0.3–4.6) | 6 (0–32)[a] | 0 (0–2.0)[b] | 0.38 |
| Albumin ratio | Median (range) | 4.0 (1.5–9.1) | 4.9 (2.1–7.0) | 4 (2.2–10) | 4 (2.7–12) | 0.60 |
| IgG index | Median (range) | 0.8 (0.4–2.7) | 0.5 (0.4–0.5) | 0.8 (0.4–3.2)[c] | 0.4 (0.3–0.5) | 0.52 |
| Oligoclonal CSF IgG bands | Yes/No | 86/6 | 0/23 | 44/2[d] | 0/18[d] | 0.72 |

CSF cerebrospinal fluid.

[a]$n$ = 46 due to missing data; [b]$n$ = 19 due to missing data; [c]$n$ = 48 due to missing data; [d]missing data exists.

*Two-sided Fisher's exact test was used for contingency tables or two-sided Mann–Whitney U test for continuous values.

**Sex was not significantly different between pwMS and HC in the discovery cohort but it was significantly different in the replication cohort ($p$ = 0.04). Age was not significantly different between pwMS and HC in either discovery or replication cohort. CSF cell count, IgG index, and oligoclonal CSF IgG bands were significantly different between pwMS and HC in both discovery and replication cohorts ($p < 0.01$).

component analysis (Fig. 2a). A Limma analysis identified 52 DEPs in the discovery cohort whereof 40 were also nominally differentially expressed ($p < 0.05$) in the replication cohort (Fig. 2b, c; see Supplementary Data 1; see Supplementary Fig. 3). Furthermore, in the replication cohort, 25 proteins were independently differentially expressed, whereof 23 proteins overlapped with the discovery cohort (Fig. 2c). Interestingly, levels of all the 52 DEPs in the discovery cohort and the 23 overlapping proteins in the replication cohort were higher in pwMS compared with controls. To investigate the MS relevance of the DEPs we performed enrichment analyses using three different sets of MS genes and proteins. We found highly significant enrichment (Fig. 2c) for genes from the DisGeNET database[24], GWAS genes[25], and known potential MS biomarkers (see Supplementary Table 1). For example, 65% of the 52 DEPs in the discovery cohort (Fisher's exact test, $p = 7*10^{-8}$) and 60% of the 25 DEPs in the replication cohort ($p = 0.002$) were associated with MS in the DisGeNET database. However, some previously suggested MS markers (including C1QA, CCL2, CXCL1, GFAP, HGF, and OPN) had non-significant log₂ fold change (FC; −0.25–0.34) when comparing pwMS to HC (see Supplementary Fig. 4). In contrast to CSF, protein profiling of plasma did not reveal any significant differences in protein expression after FDR in pwMS compared with HC (Fig. 2a, b). A few of the DEPs in CSF were also nominally differentially expressed in plasma, but with no overlap between discovery and replication cohorts (see Supplementary Fig. 5; see Supplementary Data 1). In addition, we found in general low correlation between CSF samples and plasma samples for the 52 DEPs in CSF in the discovery cohort, with the strongest correlations obtained for NfL (Pearson's correlation coefficient (PCC) = 0.46) and IL-18 (PCC = 0.33) (see Supplementary Fig. 6).

In summary, the 52 CSF proteins identified in the bigger discovery cohort represent a set of proteins being dysregulated in early stages of MS suggesting their importance in MS pathogenesis. The fact that these proteins were enriched for MS-relevant genes makes them strong biomarker candidates, and they were therefore used in the following prediction models.

**B-cell activation markers can discriminate between MS and HC**
In order to test the diagnostic potential of the 52 DEPs in CSF from the discovery cohort, we created univariate logistic regression models for each of the proteins as well as a stepwise selection model (see "Methods"). To make fair assessments of the predictive power of our inferred models we allowed no refitting of any model parameters in the

replication cohort, thus we expect the replication area under the receiver operating characteristic curve (AUC) to be a good estimation of the model test performance. In the model selection, age and sex were included as possible predictors. The highest AUC was found when having MZB1 and TNF in the model, which could predict the presence of disease with AUC = 0.99 ($p = 2*10^{-13}$) in the discovery cohort and AUC = 0.87 ($p = 6*10^{-7}$) in the replication cohort. Not surprisingly, in the univariate logistic regression models, AUC of the discovery cohort was high in all cases, but encouragingly most proteins also had high replication AUCs (Fig. 3a; see Supplementary Table 2). The top five proteins for prediction of diagnosis were MZB1, CD79B, CD27, TNFRSF13B, and IL-12p40 as ordered by AUC in the discovery cohort (Fig. 3a), where MZB1 had similar performance as the stepwise selection model containing MZB1 and TNF. These five proteins were reliably expressed above the limit of detection (LOD) in more than 95% of samples from pwMS and HC (see Supplementary Fig. 10). Finally, we investigated the discriminative power of plasma proteins. We then used the same logistic regression formulas that were trained in the CSF data of the discovery cohort and applied them to the plasma data of both cohorts. The levels of two of the derived proteins, FCN2 and IL-1RA, could discriminate pwMS from HC (AUC = 0.71 for FCN2 and AUC = 0.65 for IL-1RA) in the discovery cohort but not in the plasma data of the replication cohort. Taken together, several CSF proteins (MZB1, CD79B, CD27, TNFRSF13B, and IL-12p40) showed a strong ability to discriminate pwMS from HC, whereof the proteins MZB1, CD79B, CD27, and TNFRSF13B are related to B-cell activation.

**NfL is superior in predicting disease activity over 2 years**
Next, we aimed to create a robust model for predicting the future short-term (2-year) disease activity using the NEDA-3 concept. NEDA-3 is a binary variable based on no evidence or evidence of disease activity, as determined by reported clinical relapses, new or enlarged MRI brain lesions, or worsening in the Expanded Disability Status Scale (EDSS; see "Methods")[26]. We found that 39% of pwMS in the discovery cohort and 10% of pwMS in the replication cohort were classified as having no evidence of disease activity (NEDA) during 2 years follow-up, the remaining pwMS were classified as having evidence of disease activity (EDA). We then performed a Limma analysis of NEDA *versus* EDA groups but found no DEPs in the discovery cohort. Instead, we based the model on the 52 proteins that were differentially expressed in pwMS *versus* HC (in the discovery cohort) since these proteins were considered highly relevant to MS based on the enrichment of MS genes

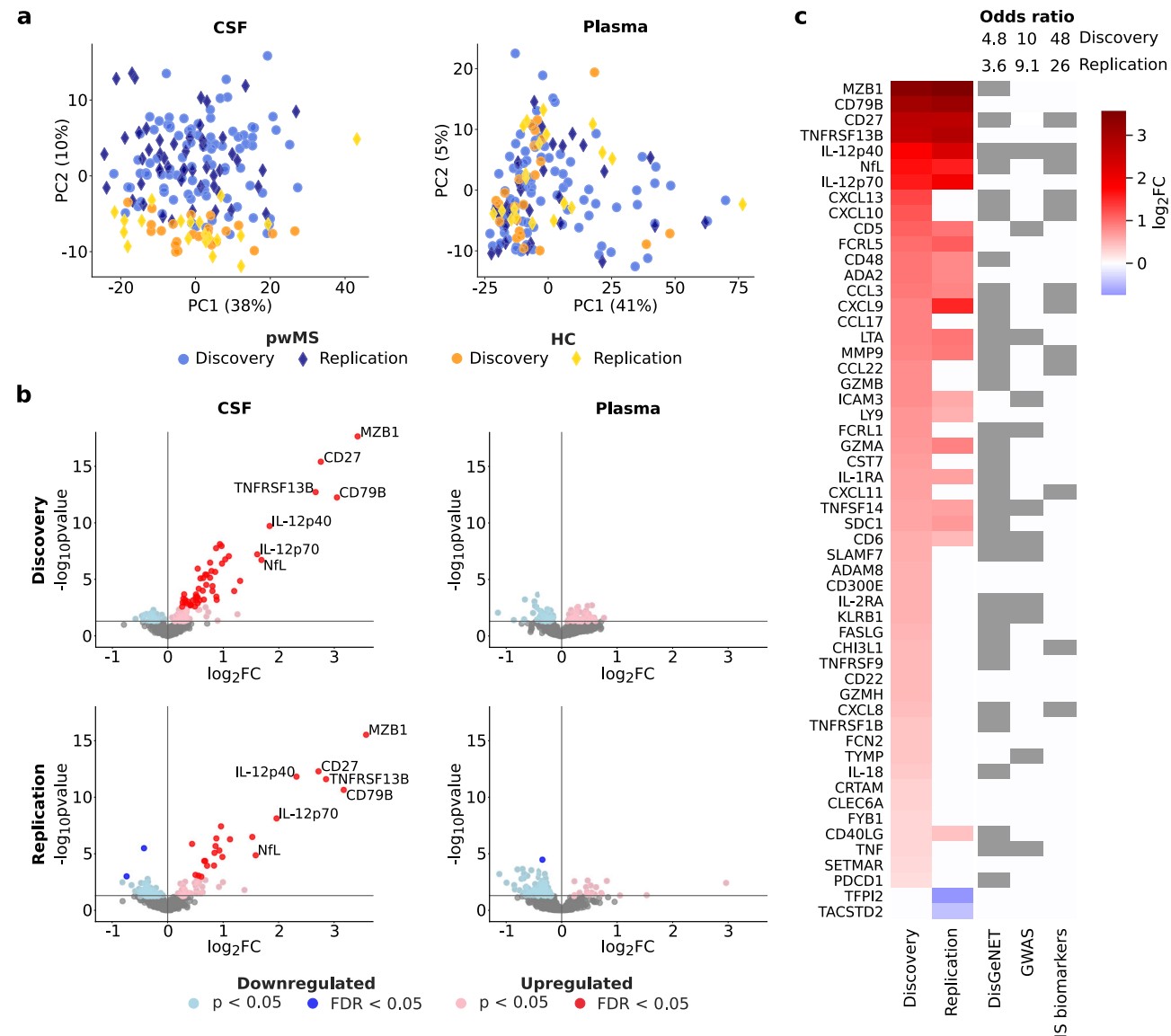

**Fig. 2 | Differential expression analysis of persons with MS (pwMS) compared to healthy controls (HC) in cerebrospinal fluid (CSF) and plasma. a** Principal component (PC) analysis of all proteins measured in the CSF samples (left) and plasma samples (right). **b** Volcano plots showing differentially expressed proteins (DEPs) in CSF (left) and plasma (right). The top upregulated proteins, which overlapped in discovery cohort and replication cohort, are marked with protein names in the plots. The differential expression analysis was performed using a two-sided linear model t-test (Limma analysis). **c** DEPs (false discovery rate < 0.05) in the CSF, in either the discovery cohort or the replication cohort. The first two columns show the log$_2$ fold change (FC) of the DEPs in each cohort. Most proteins are upregulated (red) and 23 proteins overlap in discovery and replication cohorts. In the three columns to the right, it is marked which proteins are in three different list of known MS-associated genes and proteins (DisGeNET database, GWAS genes, and MS biomarkers) with the odds ratio of the enrichment shown on the top (two-sided Fisher's exact test). The DEPs were significantly enriched for MS-associated genes from DisGeNet (discovery: $p = 7*10^{-8}$, replication: p = 0.002), GWAS (discovery: $p = 1*10^{-7}$, replication: $p = 2*10^{-4}$), and known MS biomarkers (discovery: $p = 1*10^{-12}$, replication: $p = 2*10^{-6}$).

(see above). We used a similar approach as for prediction of MS diagnosis (see above) and trained a logistic regression model for each of the 52 proteins (Supplementary Table 3) and a stepwise selection model including the 52 proteins, age, and sex as the input predictors (Fig. 3b). The best separating model was based on NfL levels in CSF and had an AUC = 0.75 ($p = 9*10^{-5}$) in the discovery cohort and an AUC = 0.77 ($p = 0.02$) in the replication cohort. In addition, IL-1RA, and CCL3 showed predictive power for disease activity, although inferior to NfL, when considering results from both the discovery and the replication cohort (Fig. 3b). A stepwise selection model (combination of NfL, IL-18, PDCD1, and CD6) showed good discrimination in the discovery cohort (AUC = 0.85) but not as good as NfL alone in the replication cohort (AUC = 0.63). In plasma we found no proteins to be of significant value to predict disease activity in either of our cohorts. Age and sex were

not selected as significant predictors in any of the models. To evaluate the potential effect of treatment, a treatment duration index covering duration and drug efficacy (first-line treatment with less effective drugs *versus* second-line treatment with more effective drugs) during the total observation time was calculated (see "Methods") and added to the models. Importantly, pwMS with EDA had in general a higher treatment duration index than pwMS with NEDA ($p = 0.02$ in the discovery cohort and $p = 0.04$ in the replication cohort, one-sided Mann–Whitney U test). Adding treatment duration index improved the predictive power of the best performing model containing only NfL (AUC = 0.77 in the discovery cohort and AUC = 0.82 in the replication cohort) but showed no significant effect on the other predictive models. The limited effect of the treatment duration index on the model performance, could partly be caused by the treatment duration

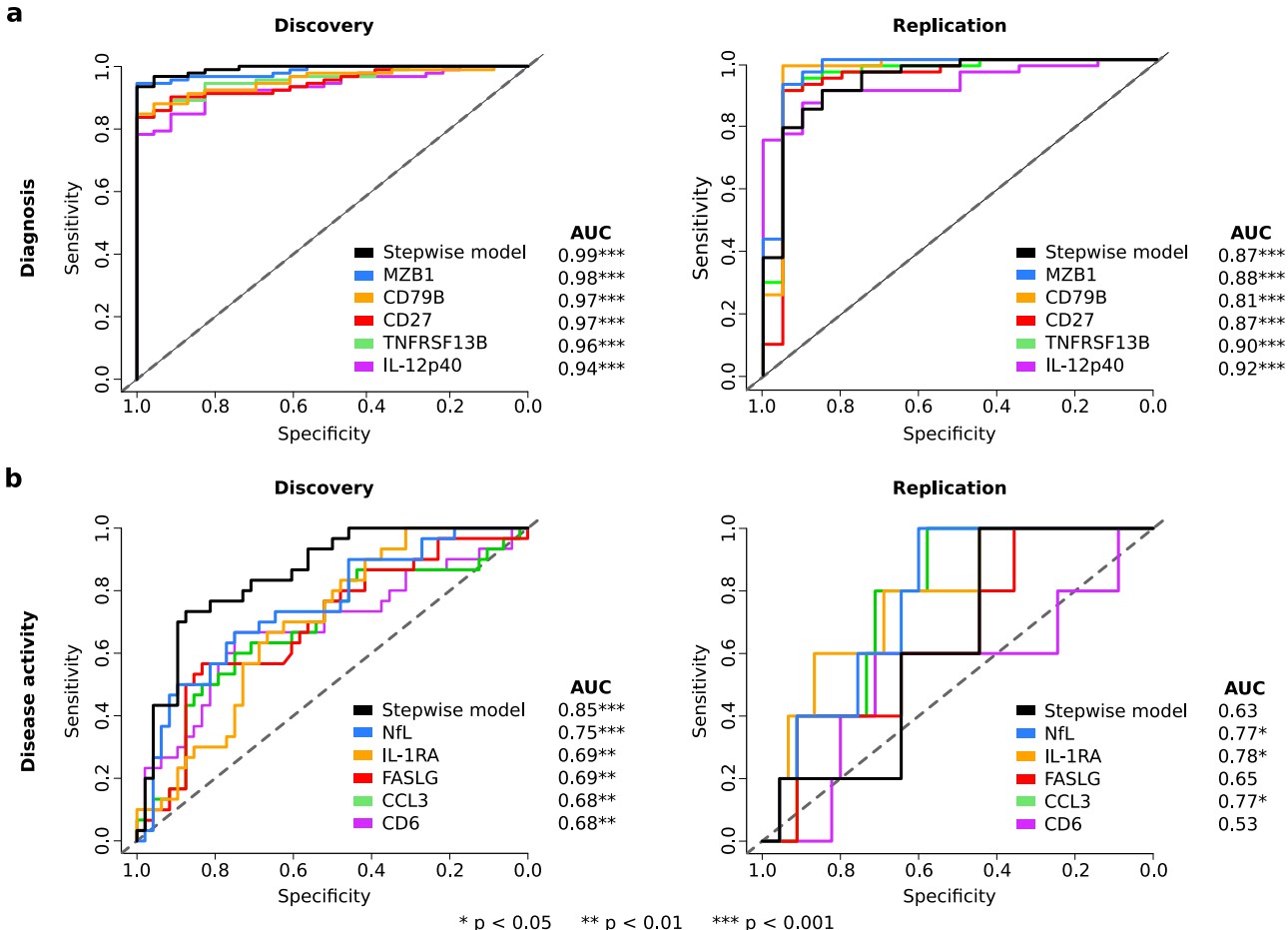

**Fig. 3 | Performance of the top cerebrospinal fluid (CSF) proteins for predicting diagnosis and disease activity over 2 years.** Predictive power, assessed by area under the curve (AUC), of the most significant CSF proteins in the discovery cohort in differentiating between **a** persons with MS (pwMS; $n = 92$ samples in the discovery and $n = 51$ samples in the replication cohort) and healthy controls (HC; $n = 23$ samples in the discovery and $n = 20$ samples in the replication cohort) and **b** pwMS showing evidence of disease activity after 2 years ($n = 48$ samples in discovery and $n = 45$ samples in replication cohort) and pwMS not showing evidence of disease activity after 2 years ($n = 30$ samples in discovery and $n = 5$ samples in replication cohort). A logistic regression model was used to assess the predictive power of both individual proteins (the top 5 proteins in the discovery cohort are

shown) and a combination of proteins, selected with a stepwise method, trained on the discovery cohort and independently validated on the replication cohort. The significance of the AUC scores were assessed with a two-sided Mann–Whitney U test. The p-values for the AUC scores of the diagnosis models in the order (stepwise model, NfL, CD79B, CD27, TNFRSF13B, IL-12p40) were ($2*10^{-13}$, $4*10^{-13}$, $1*10^{-12}$, $3*10^{-12}$, $6*10^{-12}$, $6*10^{-11}$) for the discovery cohort and ($6*10^{-7}$, $4*10^{-7}$, $2*10^{-5}$, $10*10^{-7}$, $1*10^{-7}$, $2*10^{-8}$) for the replication cohort. The p-values for the AUC scores of the disease activity models in the order (stepwise model, NfL, IL-1RA, FASLG, CCL3, CD6) were ($1*10^{-8}$, $9*10^{-5}$, 0.002, 0.003, 0.004, 0.004) for the discovery cohort and (0.19, 0.02, 0.02, 0.14, 0.03, 0.41) for the replication cohort.

index positively correlating with the expression of 34 of the 52 DEPs in the discovery cohort, although only the expression of one of these proteins (CCL3) were also significantly correlating with treatment duration index in the replication cohort (see Supplementary Fig. 7). Collectively, our findings demonstrate NfL to be the superior protein for predicting disease activity over 2 years. In addition, NfL is a very reliable marker which is expressed above the LOD in all samples from pwMS.

To facilitate the use of NfL on its own in future studies, we calculated the optimal prediction cut-off in the NfL model, and corresponding NPX level, which resulted in the maximum accuracy (see "Methods"). We found that the optimal prediction cut-off was a probability of 0.45 (accuracy = 0.71), which corresponded to an NPX level of 1.14. Using the same NPX threshold in the replication cohort resulted in an accuracy of 0.62. To translate NPX to pg/ml, we used a fraction of our data ($n = 38$) from which the NfL levels were known based on previous measurement by Simoa[27–29]. The NPX and pg/ml measurements were highly correlated (Spearman's Correlation Coefficient (SCC) = 0.97), and the NPX threshold of 1.14 corresponded to 737 pg/ml (see "Methods").

## A combination of 11 proteins accurately predicts disability worsening

Whereas the NEDA-3 concept reflects the short-term disease activity mainly by detecting relapses and MRI activity, the long-term disability progression is more relevant from the perspective of a person with MS since it directly affects the quality of life[30]. The EDSS is the most used measure of disability status, but to adjust for age, the age-related MS score (ARMSS) was created[31]. To further adjust for length of observation time and allow for using data from different lengths of follow-up time, we used the recently described normalized ARMSS (nARMSS; see "Methods"). To obtain an nARMSS score, a person had to have had at least two documented EDSS scores over a period of at least 3 years. The resulting cohorts used for predictions consisted of 71 pwMS in the discovery cohort and 33 pwMS in the replication cohort. In Fig. 4, each person's EDSS scores for each follow-up year and the resulting nARMSS score are shown and described in further detail in Supplementary Fig. 8. The nARMSS scores can obtain a value between −5 and +5, where a score of 0 represents the average disability worsening of pwMS based on historical cohorts ($n = 25,558$)[31]. Both the discovery and replication cohorts showed an overrepresentation of pwMS with a

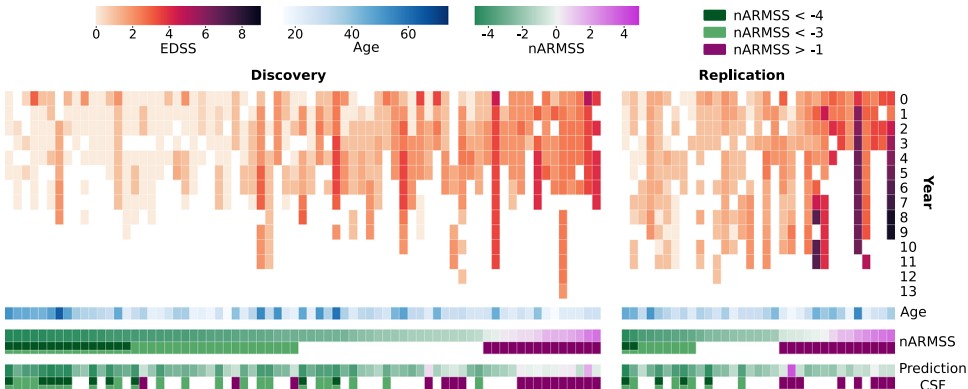

**Fig. 4 | Overview of the Expanded Disability Status Scale (EDSS) scores during yearly follow-up for persons with MS (pwMS).** The disability worsening scores for pwMS, who had at least two EDSS scores over a period of more than 3 years. Each column corresponds to one person. The top heatmap shows the EDSS scores for each follow-up year (0–13 years), followed by the age of each person. White cells indicate that no EDSS score was available for that year. Thereafter follows the normalized age-related MS score (nARMSS), calculated from a person's EDSS score and age. In the row underneath the nARMSS score it is marked if a person's nARMSS score is below the thresholds nARMSS < −4 or nARMSS < −3, or above the threshold nARMSS > −1. White cells indicate that the nARMSS score is not covered by any of these three thresholds. The last two rows show the predicted nARMSS score obtained from the suggested cerebrospinal fluid (CSF) model combining 11 proteins (first row) and if the predicted nARMSS score is covered by any of the three thresholds mentioned above (second row).

less severe disability worsening, with 50% of the pwMS having a score below −3.0 in the discovery cohort and below −2.0 in the replication cohort (see Supplementary Fig. 9). The nARMSS scores had a significantly stronger correlation with the last ARMSS score (age adjusted EDSS) compared to the first ARMSS score, used for calculating nARMSS, for both the discovery cohort (SCC = 0.89 compared to SCC = 0.71, $p = 0.003$) and the replication cohort (SCC = 0.92 compared to SCC = 0.79, $p = 0.03$). We first tested if short-term disease activity (based on 2-year NEDA-3) was associated with nARMSS but found no significant difference in nARMSS when comparing EDA ($n = 43$) with NEDA ($n = 27$; medians were −2.90 and −3.36, respectively, two-sided Mann–Whitney U-test $p = 0.15$). Then we also tested and found that age at baseline and subsequent treatment (treatment duration index) were correlating with nARMSS with an SCC = 0.38 ($p = 0.001$) and an SCC = 0.28 ($p = 0.02$), respectively, which led us to further include them as possible covariates in our models in downstream analysis.

To create a predictive model of nARMSS, we first performed a Limma analysis of the 1463 proteins based on the nARMSS score, but no DEPs were identified. Therefore, we again started from the 52 DEPs in CSF of pwMS compared to HC in the discovery cohort (see above), age, and sex. The predictive model of nARMSS was performed with a stepwise linear regression model using the CSF protein data. This resulted in a significant model including eleven proteins (CXCL13, LTA, FCN2, ICAM3, LY9, SLAMF7, TYMP, CHI3L1, FYB1, TNFRSF1B, NfL) and age as predictors (see Supplementary Table 4). We also evaluated the effect of treatment, by adding treatment duration index to the model, but it did not improve the performance of the model. The model consisted of both proteins with positive and negative coefficients, even though all proteins were upregulated in MS compared to HC. Next, when comparing the predicted nARMSS with the true nARMSS we found strong and significant correlations in both the discovery (SCC = 0.69, $p = 3*10^{-11}$) and the replication cohort (SCC = 0.74, $p = 9*10^{-7}$; Fig. 5a). To also consider both the correlation and accuracy of the prediction, we used Lin's concordance correlation coefficient (CCC) as an additional performance metric, which resulted in a CCC of 0.72 ($p = 2*10^{-12}$) in the discovery cohort and a CCC of 0.51 ($p = 0.002$) in the replication cohort. As a comparison, we also evaluated the performance of models only including age and each of the 11 proteins and found that the combined model outperformed each of the individual models (see Supplementary Table 4).

To further evaluate the performance of the model, we assessed the ability to predict groups of pwMS with similar disability worsening. We made three different divisions using three different nARMSS thresholds, selected using the discovery cohort: nARMSS < −4 (corresponding to 20% of pwMS with the best prognosis), nARMSS < −3 (corresponding to 50% of the pwMS, i.e., a median split), and nARMSS > −1 (corresponding to 20% of the pwMS with the worst prognosis). For each of these thresholds, the model successfully identified the selected pwMS group both in the discovery and the replication cohort. For each respective threshold the AUC for the discovery cohort was 0.85 ($p = 2*10^{-5}$), 0.76 ($p = 7*10^{-5}$), and 0.92 ($p = 6*10^{-7}$) with an accuracy of 0.85, 0.66, and 0.85 and the AUC for the replication cohort was 0.90 ($p = 0.03$), 0.88 ($p = 4*10^{-4}$), and 0.90 ($p = 6*10^{-5}$) with an accuracy of 0.88, 0.85, and 0.82 (Fig. 5b). Lastly, we confirmed that the 11 identified proteins were reliably expressed above the LOD in more than 60% of samples from pwMS whereof eight proteins were expressed in more than 75% of samples from pwMS (See Supplementary Fig. 10). The performance of models with the three proteins (SLAMF7, TYMP, FYB1) removed which did not fulfill the more stringent threshold of 75% can be seen in Supplementary Table 5.

We continued by investigating the potential of the model to predict nARMSS from plasma samples. Interestingly, the model was enriched ($p = 0.03$) for proteins whose expression in CSF correlated with the expression in plasma ($p < 0.05$ in the discovery cohort). Of the 52 DEPs in CSF, seven proteins had correlating expressions in CSF and plasma, whereof four were selected in the model: NfL (SCC = 0.45), CXCL13 (SCC = 0.30), CHI3L1 (SCC = 0.27), and FCN2 (SCC = 0.25; see Supplementary Table 4). We hypothesized that the correlating proteins could be used to predict nARMSS from plasma samples by using a model trained on CSF samples. Again, performing a stepwise linear regression model, only selecting among the four correlating proteins and age, we reduced the model to three terms: intercept (coefficient (c) = −0.707), age (c = −0.068) and NfL (c = 0.369). The model could predict nARMSS from plasma samples with an SCC of 0.40 ($p = 5*10^{-4}$) and a CCC of 0.28 ($p = 0.02$) in the discovery cohort ($n = 71$), and an SCC of 0.60 ($p = 0.04$) and a CCC of 0.14 ($p = 0.66$) in the replication cohort ($n = 12$, Fig. 5c). Evaluating the model based on the three nARMSS thresholds (nARMSS < −4, nARMSS <−3, nARMSS >−1) resulted in discovery AUC of 0.78 ($p = 4*10^{-4}$), 0.59 ($p = 0.09$), and 0.74 ($p = 0.003$), with an accuracy of 0.77, 0.56, and 0.82 and replication AUC of 1.0 ($p = 0.08$), 0.70 ($p = 0.19$), and 0.78 ($p = 0.07$) with an accuracy of 1.0, 0.58, and 0.50 (Fig. 5d). It should be noted that only 12

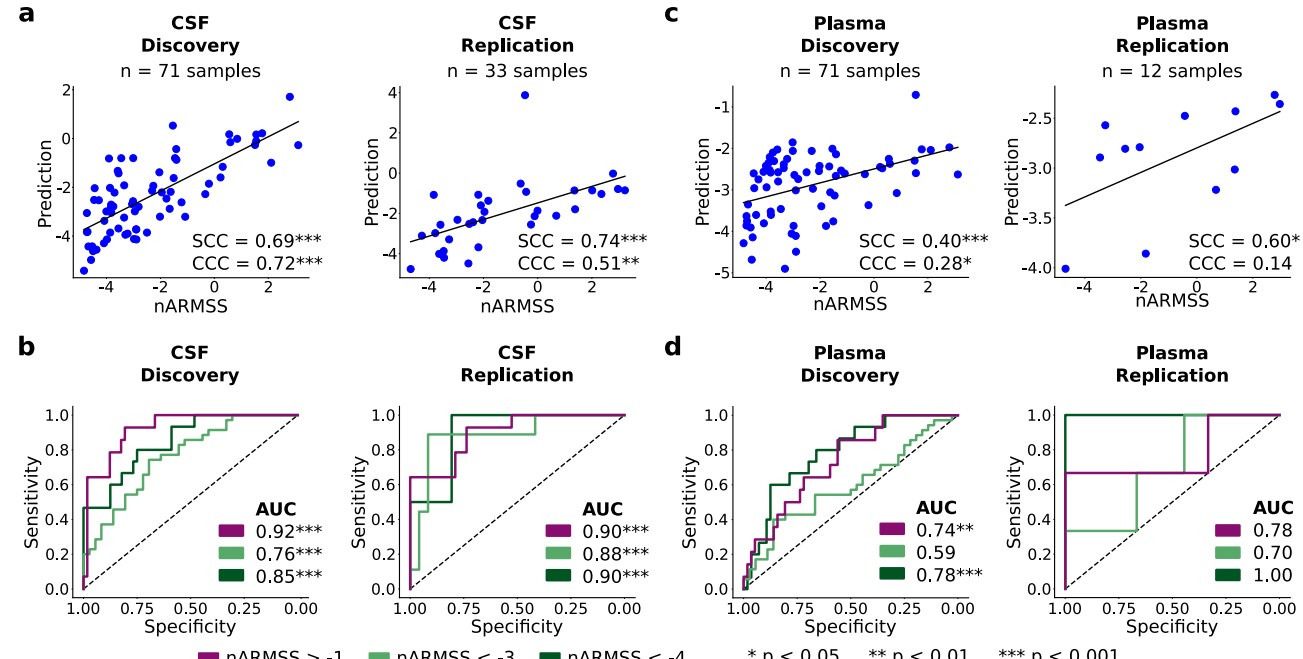

**Fig. 5 | Performance of the top models for predicting long-term disability worsening using cerebrospinal fluid (CSF) and plasma proteins. a** CSF: The predicted normalized age-related MS scores (nARMSS) were significantly correlating with the true nARMSS for both discovery and replication cohorts, assessed with Spearman's correlation coefficient (SCC; discovery: $p = 3*10^{-11}$, replication: $p = 9*10^{-7}$) and Lin's concordance correlation coefficient (CCC; discovery: $p = 2*10^{-12}$, replication: $p = 0.002$). **b** CSF: Receiver operating characteristic (ROC) curves and area under the curve (AUC) scores for each of the three different nARMSS thresholds. The p-values for the AUC scores in the order (nARMSS > −1, nARMSS < −3, nARMSS < −4) were ($2*10^{-5}$, $7*10^{5}$, $6*10^{-7}$) for the discovery cohort and (0.03, $4*10^{-4}$, $6*10^{-5}$) for the replication cohort. **c** Plasma: Reducing the CSF model to NfL and age resulted in a model that could predict nARMSS from plasma samples. The predicted nARMSS significantly correlated with the true nARMSS for both the discovery cohort (SCC: $p = 5*10^{-4}$, CCC: $p = 0.02$) and replication cohort (SCC: $p = 0.04$, CCC: $p = 0.66$). **d** Plasma: ROC curves and AUC scores for each of the three different nARMSS thresholds. The $p$-values for the AUC scores in the order (nARMSS > −1, nARMSS < −3, nARMSS < −4) were ($4*10^{-4}$, 0.09, 0.003) for the discovery cohort and (0.08, 0.19, 0.07) for the replication cohort. The significance of the SCCs and CCCs was assessed with t-statistics (two-sided) and the significance of the AUC scores were assessed with a one-sided Mann–Whitney U test.

pwMS in the replication cohort had both usable plasma samples and fulfilled the requirements for obtaining an nARMSS score.

## Network analysis provides functional context for DEPs and reveals additional biomarker candidates

To provide a functional context of the discovered MS proteins we made an MS network using STRING version 11.5[32]. The 11 proteins in the nARMSS model and the 23 DEPs that overlapped in the discovery and the replication cohort, representing a set of core proteins in MS, were connected by adding at most one intermediate protein. The proteins, except ADA2, formed a closely connected network consisting of 40 proteins, including 11 intermediate proteins (Fig. 6a, Supplementary Fig. 11a). Among the intermediate (added) proteins there were five proteins that were not included in the proteomics profiling; the chemokine receptors CCR1 and CCR5, the receptor ITGAL expressed on leukocytes, the adapter protein LCP2 associated with the T-cell receptor, and the multifunctional adapter protein SDCBP. The resulting MS network had 13.5 times as many interactions than is expected ($p < 1*10^{-16}$) using the STRING protein–protein interaction network, indicating shared biological functionality[32]. Gene Ontology enrichment analysis showed that the MS network was highly enriched for proteins involved in cytokine-mediated signaling ($n = 11$, $p = 7*10^{-7}$), T-cell activation ($n = 14$, $p = 3*10^{-9}$) and B-cell activation ($n = 6$, $p = 6*10^{-4}$), exocytosis ($n = 4$, $p = 0.03$) and endocytosis, in particular phagocytosis (n = 4, $p = 0.01$), cell adhesion including regulation of cell-cell adhesion and cell-cell adhesion via plasma-membrane adhesion molecules ($n = 11$, p = 0.02), apoptotic processes including positive regulation of apoptotic process ($n = 9$, $p = 6*10^{-5}$) and negative regulation of leukocyte apoptotic process ($n = 2$, $p = 3*10^{-2}$), myelination including regulation of myelination ($n = 2$, $p = 0.02$). Some

proteins in the network were not annotated by Gene Ontology and were therefore manually categorized based on the literature (Fig. 6b; see references in Supplementary Table 6). In addition, we performed a KEGG pathway enrichment analysis and found enrichment for pathways such as cytokine-cytokine receptor interaction ($p = 2*10^{-14}$) and cell adhesion molecules ($p = 9*10^{-4}$; see Supplementary Fig. 11b). Lastly, we investigated the MS enrichment of the 11 intermediate proteins and found high enrichment of MS genes from both DisGeNET (odds ratio = 29.1, $p = 6*10^{-8}$) and GWAS (odds ratio = 14.2, $p = 0.002$), with 8 of the intermediate proteins associated to MS in the DisGeNET database[24].

## Discussion

Early prediction of prognosis in MS is a key factor for optimizing therapeutic management and benefit-risk balance. Here we took advantage of a newly developed highly sensitive and robust PEA technique to perform data-driven testing of 1463 proteins in CSF and plasma of 186 individuals to find accurate signatures for short- and long-term prognosis in early MS. In CSF, but not in plasma, we observed a clear separation between early MS and HC by identifying a signature containing 52 DEPs that were enriched for MS-relevant proteins based on previous GWAS and biomarker studies. When testing these early upstream CSF proteins independently and in combinations for prognostic ability, a set of 11 proteins in CSF were able to accurately predict long-term disability as measured by nARMSS and based on an average of 6 years follow-up in both a discovery and a replication cohort. In plasma, only NfL was able to predict nARMSS with moderate accuracy. For prediction of short-term disease activity based on 2-year NEDA-3, only CSF levels of NfL showed a high accuracy in both cohorts. Of note, we consistently used the same pwMS cohorts from two

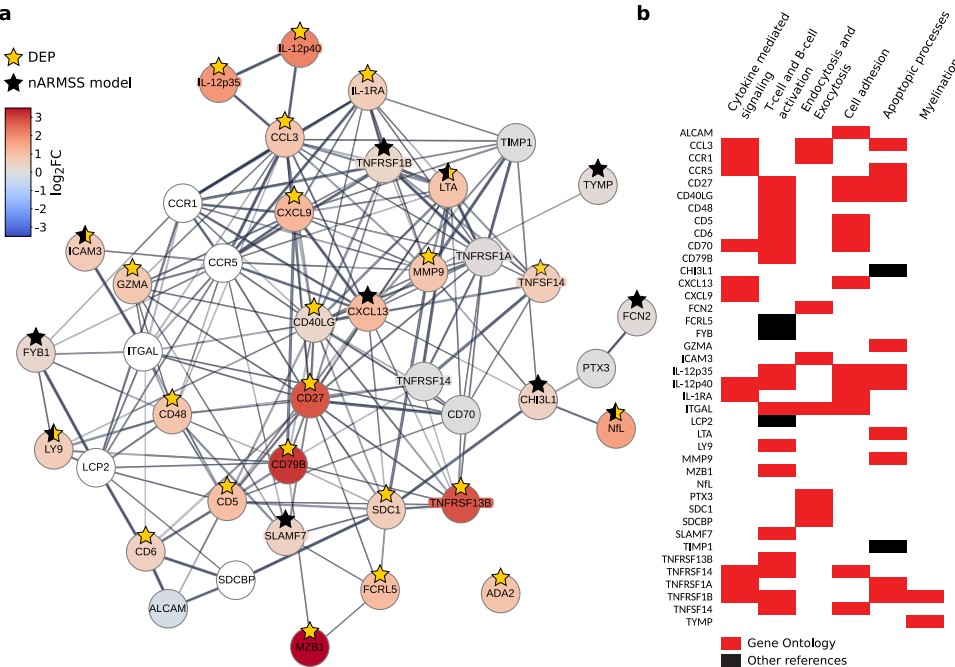

**Fig. 6 | Identified MS proteins share functional context. a** An MS network was formed by connecting the proteins in the normalized age-related MS score (nARMSS) model (black star) and the differentially expressed proteins (DEPs) that overlapped in the discovery and the replication cohort (yellow star). The proteins were connected using STRING (combined interaction score > 0.4) with one inter-mediate protein allowed to be added to connect proteins. The proteins are color-coded on the log₂ fold change (FC), comparing persons with MS with healthy controls. The white colored proteins were not included in the proteomics profiling. IL-12p35 was not measured in the proteomics profiling but is included as a DEP as it together with IL-12p40 represents IL-12p70. The linewidth of the interactions is related to the combined interaction score of each interaction in STRING. **b** Proteins in the MS network divided into functional categories. The proteins were categorized into groups with shared functionality based on Gene Ontology enrichment analyses (red), and the literature (black).

different sites and allowed no retraining of any prediction model parameters in the replication cohort, thus increasing the generalizability and ability for successful replication of models also in other cohorts. Collectively, our study reveals several proteins relevant in MS pathogenesis as well as demonstrates a set of proteins important for prediction of disability outcome. In addition, NfL was confirmed as a robust marker of short-term disease activity.

A major finding of the present study was the ability to predict long-term disability, as based on EDSS during an average follow-up of 6 years. To ensure that EDSS scores were comparable across studies, we utilized nARMSS, a score that not only takes into account disease duration and age[31,33], but also allows for the incorporation of EDSS data from various time points and follow-up periods. This feature enables the comparison of disability progression and outcomes across cohorts with varying levels of data density[34]. Although nARMSS is intended to account for age, we observed an overcorrection for age, and consequently, we found age to be an important factor to include in our nARMSS model. It is increasingly recognized that disability over time may occur independent of relapse-associated inflammatory activity[35]. This notion is also supported by our finding of no significant correlation between EDA and nARMSS. Hence, it is of crucial importance to study disability development by itself and to find markers for predicting disability progression unrelated to overt inflammation.

Previous studies on prediction of disability (based on EDSS) are limited by a short follow-up time[36], few included parameters, such as CSF lymphocyte count[37] and clinical measures[36], or using expression levels of a limited number of proteins[38–40]. Sufficient follow-up time is necessary since it usually requires several years for pwMS to display changes in their disability scores[41]. Although some studies have suggested NfL as a potential biomarker for disability prediction[38,40], they lacked a replication group. Importantly, in our study including a replication cohort, the suggested model could identify both pwMS with a low and high chance of developing a high nARMSS score, thus

providing a promising predictive tool for identifying clinical course at different ends of the heterogeneous disease spectrum of MS. Since early treatment has proven long-term benefits regarding disability outcome despite a seemingly mild disease initially[15,42,43], a biomarker signaling an increased risk of long-term disability progression detectable in early disease stages would strengthen a prompt high-efficacy treatment at first diagnosis of MS. Thus, if further confirmed, our identified set of proteins would be of value as biomarkers in individualized treatment protocols to avoid both over- and under-treatment in terms of drug efficacy, which is highly relevant with respect to risk of side-effects as well as costs.

Several of the suggested proteins in our model for predicting nARMSS have been validated in previous studies for their clinical relevance in MS. Some of these proteins such as CXCL13[44–46], LTA[46], SLAMF7[47], CHI3L1[44], and NfL[45] are well-established as valuable markers for prognostic assessment and treatment response[48] while the proteins TNFRSF1B, FCN2, ICAM3, LY9, TYMP[47], and FYB1 are less represented in the literature. NfL level in CSF is an established marker of ongoing neuroaxonal damage in MS, with emerging data supporting its usefulness also in the blood compartment. Furthermore, levels of NfL in both CSF and plasma/serum are also shown to decrease with disease modifying treatments[49–51] and CSF-NfL is able to predict short-term disease activity manifested by contrast-enhancing lesions, relapses, or both[27,45,46,52–54]. Despite covering 1463 proteins in our present study, CSF-NfL stood out as the major biomarker for prediction of short-term disease activity. Since brain-derived NfL can leak out to the circulation, it can be measured in plasma with highly sensitive methods, revealing fairly good correlations between CSF and plasma[28,55]. Thus, plasma or serum NfL has been suggested as an easy-accessible emerging biomarker, although not yet proven[19,28,50,56–60]. However, in our study, we found no reliable biomarker candidates in plasma regarding short-term disease activity. Interestingly, however, we found that CSF-NfL levels alone could not predict long-term disability worsening as

measured by nARMSS. At the same time, we found NfL in plasma to be a marker for disability worsening (nARMSS), although a combination of CSF proteins including NfL showed a substantially higher accuracy. NfL in plasma showing stronger predictive power than CSF-NfL is surprising but is corroborating a recent finding that serum NfL has stronger correlation with MS severity outcomes than CSF-NfL[61].

It is difficult to draw any conclusions on the effect of treatment on disease activity or disability progression based on our study due to its observational nature which means that the assignment of treatments to patients was not randomized but based on clinical judgement. Our inclusion of the treatment duration index as a feature in the models was not aimed at establishing a direct causal relationship between treatment and outcomes but rather to offer a comprehensive representation of the individuals' clinical profiles. The decision whether the treatment duration index remains in the model or not is determined by the machine learning process, which relies on the correlations between features. The significant correlations between treatment duration index and several proteins in our nARMSS model might explain why treatment duration index did not improve the final model. Achieving a good performance on both the discovery and replication cohort, despite none of the selected proteins significantly correlating with treatment duration index in both cohorts, suggests that these proteins have strong predictive power for MS progression.

This is the first study in MS utilizing the sensitive PEA technology combined with next-generation mass sequencing (PEA-NGS), which allows for simultaneous detection of nearly 1500 proteins. The PEA-NGS is developed from PEA-qPCR, which is limited to smaller panels of targeted proteins, and the two methods have shown excellent correlations in targeted panels ($n = 384$ proteins)[5]. We recently showed promising results of using PEA-qPCR for robust detection of a 92-protein inflammation-panel in CSF and plasma of MS and controls[10]. We here confirm that IL-12p40, IL-12p70, CXCL9, CD5, MMP9, and NfL again showed diagnostic power for differentiating pwMS and HC. The top proteins (MZB1, CD79B, CD27, and TNFRSF13B) in our study with the ability to discriminate MS from HC are all expressed in B cells and have been associated with several chronic autoimmune diseases, including MS[62,63]. While our study does not address the question of whether these proteins can differentiate MS from other neurological diseases (ONDs), it does shed light on the significance of B cell activation in MS pathology. This is in line with another study on CSF biomarker-based diagnostic tools in which other proteins related to expansion and activation of B cell/plasma cell lineages were shown to effectively distinguish MS from ONDs[64]. Whether the proteins identified in our study will prove useful for differentiating MS from ONDs remains to be settled.

By analyzing the network of core proteins, based on the predictive proteins and the DEPs in both the discovery and the replication cohort, we found most MS proteins to be functioning in a densely connected network. The proteins in the network were mostly associated with the immune response, with proteins supporting that both T cells and B cells are central in the pathogenetic process of MS[65], for example, the B cell chemoattractant CXCL13, and the Th1 cell chemoattractant CXCL9. Furthermore, CD27 and CD70 play a role in a costimulatory process that allows B cells to maintain activation of pathogenic T cells[66]. Moreover, the network shows the importance of MZB1, which may be involved in MS pathology through activating autoproliferative CD4[+] T cells and pathogenic B cells in CSF and is thought to potentially trigger B cell response against Epstein–Barr virus (EBV) proteins[67]. Among the intermediate proteins, we identified a group of proteins that were not included in our initial protein panels (CCR1, CCR5, ITGAL, LCP2, SDCBP), whereof the proteins ITGAL, LCP2, and SDCBP can be detected in blood by mass spectrometry[68]. We propose all these proteins mentioned as potential MS biomarkers to be validated in future studies.

Our study comes with limitations. In addition to pwMS, a group of people with ONDs would be highly relevant to see if any of these biomarkers can distinguish between MS and ONDs. Such a group, or groups with ONDs, were not included in the present study since our focus was to predict disease course rather than diagnosis of MS. The long follow-up period of up to 13 years is a strength of the study, but not all pwMS had this long follow-up time. To address this limitation, we utilized nARMSS scores, which account for varying follow-up durations. However, the accuracy of the nARMSS score evidently improves with longer follow-up periods and frequent EDSS assessments. Since we had long follow-up times in our study, many pwMS were taking different medications throughout the observation time. This is a challenge for including the treatment as a covariate when building the models. Hence, we recognize that the treatment duration index that we used in our study is an attempt to simplify a more complex effect and may not reflect the full picture.

In conclusion, we identified several promising protein biomarkers which could be used to predict short-term activity and long-term disease progression in newly diagnosed MS. This is useful for aiding personalized treatment strategies, to both reduce costs and side effects of current treatments.

## Methods
### Study design and sample handling
People with clinically isolated syndrome (CIS) or RRMS were enrolled in a prospective longitudinal cohort study from two sites. CSF and plasma samples were taken from 92 people with CIS or RRMS at the Department of Neurology, Linköping University Hospital, Sweden and 51 people with CIS or RRMS at the Karolinska University Hospital, Sweden. Everyone fulfilled the revised McDonald criteria from 2010 and 2017[69,70] for CIS or MS. Peripheral blood and CSF were sampled from everyone at baseline. pwMS underwent clinical neurological examination including EDSS, and MRI at baseline and at several time points afterward as follow-up. During the study, pwMS received immunomodulatory treatment according to Swedish national and local clinical praxis. Age-matched HC were recruited from healthy blood donors (23 at the Linköping University Hospital and 20 at the Karolinska University Hospital). HC from Linköping University Hospital were also sex-matched. HC had no past or current neurological and autoimmune disease, and their clinical neurological examinations were normal as were routine findings in CSF. Peripheral blood and CSF were sampled from all HC. No medication, except oral contraceptive pills, was allowed in HC. Sex of pwMS and HC were determined based on information provided in Swedish official medical records. Demographic data and clinical data are presented in Table 1 and Table 2, respectively. Clinical data for each person with MS is available in Supplementary Data 2. If there was a significant difference between the two cohort for the characteristics presented in Tables 1 and 2 was assessed using two-sided Fisher's exact test (fisher_exact from the python package SciPy v 1.9.1) for contingency tables or two-sided Mann–Whitney U test (mannwhitneyu from the python package SciPy) for continues values.

Plasma and CSF samples were collected from all pwMS and HC at both sites. For the discovery cohort (Linköping University Hospital): Blood was collected in EDTA tubes (BD Vacutainer®, Beckton Dickinson, Franklin Lakes, NJ, US) and centrifuged at $1500 \times g$ for 10 min in room temperature (RT) within 2 h from sampling. The plasma was aliquoted and stored at −70 °C. The CSF was kept cold after sampling and processed within one hour by centrifugation $300 \times g$ for 10 min in RT to pellet and remove cells. The supernatant was aliquoted and immediately frozen and stored at −70 °C. For the replication cohort (Karolinska University Hospital): Blood was collected in EDTA tubes (BD Vacutainer®, Beckton Dickinson) and centrifuged at $1700 \times g$ for 15 min in RT. The CSF was centrifuged at $350 \times g$ for 12 min in RT. Both

**Table 2 | Clinical data of persons with MS**

| | | Discovery cohort | Replication cohort | p-value∗ |
|---|---|---|---|---|
| Cohort size | n | 92 | 51 | NA |
| Diagnosed with RRMS at baseline | n | 30 | 30 | **0.003** |
| Diagnosed with CIS at baseline | n | 62 | 21 | **0.003** |
| Disease duration before baseline (months) | Median (range) | 4 (0–136) | 4 (0–128) | 0.61 |
| Relapse within one month before baseline | Yes/No | 18/74 | 11/40[a] | 0.83 |
| No. of relapses within 2 years before baseline | Median (range) | 1 (0–3) | 1 (1–5) | 0.12 |
| No. of T2 lesions at baseline MRI | 0 | 10 | 1 | 0.12 |
| | 1–9 | 66 | 24 | **0.004** |
| | 10–20 | 3 | 11 | **8∗10⁻⁴** |
| | >20 | 13 | 15 | **0.04** |
| No. of MRI Gd+ lesions at baseline | Median (range) | 0 (0–10) | 0 [b] (0–5) | 0.29 |
| EDSS at baseline | Median (range) | 1 (0–4.5) | 1.5 (0–3.5) | **0.002** |
| Treatment within 3 months before baseline | Yes/No | 5/87 | 0/51 | 0.16 |
| | Type | NZB (n = 2), RTX (n = 2), DMF (n = 1) | NA | NA |
| Steroid treatment within 3 months before baseline | Yes/No | 9/83 | 2/49 | 0.33 |
| Observation time after baseline (years) | Median (range) | 5.4 (0–12.6) | 3.1 (0.5–11.5) | 0.61 |
| Observation time >3 years | n | 71 | 33 | 0.59 |
| Treatment duration index (TDI) | Median (range) | 0.47 (0–1.69) | 0.97[a] (0–1.85) | **7∗10⁻⁸** |
| No. with TDI < 50% | n | 66 | 20 | **3∗10⁻⁴** |
| No. with TDI 50–90% | n | 24 | 29 | **3∗10⁻⁴** |
| No. with TDI > 90% | n | 2 | 1 | 1.0 |
| NEDA over 2 years | Yes/No | 30/48 | 5/45 | **5∗10⁻⁴** |
| EDA caused by MRI | n | 27 | 38 | 0.23 |
| EDA caused by EDSS | n | 7 | 0 | **0.01** |
| EDA caused by Relapse | n | 14 | 7 | 0.14 |

CSF cerebrospinal fluid, RRMS relapsing remitting multiple sclerosis, CIS clinically isolated syndrome, EDSS Expanded Disability Status Scale, DMF Dimethyl Fumarate, EDA evidence of disease activity, Gd+ Gadolinium enhanced, NEDA no evidence of disease activity, NZB Natalizumab, RTX Rituximab.
[a]Missing data exists; [b]n = 39 due to missing data.
∗Two-sided Fisher's exact test was used for contingency tables or two-sided Mann–Whitney U test for continuous values. Bold p-values are below 0.05.

plasma and CSF samples were prepared within 2 h of sampling, and all were stored at −80 °C immediately after handling. All included samples were thawed on ice and transferred to 96-well plates for further analysis at the SciLifeLab Biomarker facility. The samples from each site and cohort were randomly distributed on the plates to minimize potential batch effects between sites and sample groups.

**Proteomics profiling and data pre-processing**
The concentrations of 1463 proteins was measured using the Olink Explore platform which uses PEA technology. The proteins were pre-selected from four Olink panels: Explore 384 Cardiometabolic, Explore 384 Inflammation, Explore 384 Neurology, and Explore 384 Oncology. In the Olink Explore platform, massive parallel sequencing is used instead of qPCR in the previous target panels[6]. The protein concentrations are given as Olink's relative protein quantification unit on $\log_2$ scale: Normalized Protein Expression (NPX). The NPX values were intensity normalized by Olink[7]. The plasma samples from one pwMS subcohort (n = 21) in the replication cohort had significantly higher protein concentrations than the remaining plasma samples. We suspected that the difference could be caused by sampling handling variability and attempted to correct for the difference in protein concentration using the approach described by[23]. However, the attempted correction was not satisfying, and the 21 plasma samples were therefore removed from further analysis. The CSF data from these 21 individuals did not differ from other CSF samples and were therefore used. The data was further pre-processed by removing proteins with NPX below the LOD in more than 75% of the samples, resulting in 1009 proteins in the CSF samples and 1367 proteins in the

plasma samples. For the remaining proteins with NPX values below the LOD in some samples, the reported NPX values were kept in the data unchanged. In addition, we confirmed that removing proteins below the LOD, based on all samples, did not exclude any valuable protein markers with an unbalanced distribution of values below the LOD in the different groups (pwMS and HC; see supplementary Fig. 12). The mean expression was used for proteins that had been measured in several panels. Before using the data for developing predictive models, we checked for batch effect using singular value decomposition analysis (see Supplementary Fig. 13). Although no prominent batch effects were noted, the data was corrected in two steps. First, the protein levels were corrected so that the controls in the discovery cohort and the replication cohort had the same mean and standard deviation. Second, we applied the batch correction method ComBat using the function runCombat from the R-package ChAMP (v2.21.1)[71].

**Differential expression analysis**
Differential expression analysis was performed using the R-package Limma (v3.52.4)[72]. A linear model was fitted to the data before empirical Bayes moderated t-statistics were calculated and multiple testing correction (Benjamin-Hochberg) was performed. The threshold FDR < 0.05 was used to determine if a protein was differentially expressed. For all analysis, except for disease duration at baseline sampling and nARMSS, the comparison was made between two groups. For disease duration and nARMSS, the comparison was made on the continuous values with age included as covariate for the linear model fitting. In the differential expression analysis $\log_2$FC values for all proteins were also obtained.

## Enrichment analysis of MS-associated proteins

Enrichment of MS-associated proteins was assessed using two-sided Fisher's exact test (fisher_exact) from the python package SciPy (v1.9.1)[73]. Three different lists of MS-associated proteins were used:

(1) DisGeNET: Genes associated to MS (C0026769; $n = 1800$ genes) were downloaded from the DisGeNET database v7.0[24].

(2) GWAS: MS SNPs from GWAS ($p < 1*10^{-6}$) were obtained from ref. 25 and mapped to the closest gene ($n = 573$ genes).

(3) MS biomarkers: a list of known MS biomarkers ($n = 19$ biomarkers) was compiled (references in Supplementary Table 1). Only proteins measured in the proteomics profiling were considered for inclusion among the known MS biomarkers.

## NEDA-3 concept

NEDA-3 is an established way of evaluating the absence of disease activity in MS[26] based on three notions; (1) no clinical relapses; (2) no progression in the EDSS; or (3) no new lesions or enlarged lesions showed by MRI, resulting in a binary outcome of showing EDA or showing NEDA. The assessment is regularly performed by a neurologist. A progression in EDSS score was determined based on: EDSS increase of 1.5 if baseline EDSS = 0, EDSS increase of 1 if baseline EDSS ≥ 1, and EDSS increase of 0.5 if baseline EDSS > 5. In our study the outcome of NEDA-3 assessment for each person during 2 years follow-up (±6 months) from the sampling time was used in the logistic regression modeling.

## Treatments and treatment duration index

MS treatments were categorized into two main groups: first-line or less effective treatments such as interferon beta-1a, copaxone, human normal immunoglobulin (IVIg), dimethyl fumarate, teriflunomide, Solu-Medrol, laquinimod and second-line or more effective treatments such as rituximab, natalizumab, fingolimod, cladribine, siponimod, daclizumab, Hematopoietic stem cell transplantation, ofatumumab, ocrelizumab, mitoxantrone (references used to categorize these treatments into two groups can be found in Supplementary Table 7). Since the efficacy and duration of treatment affect the long-term disability outcome, we calculated the proportion of the observation time during which the pwMS were on second-line treatments (including the period before the study was initiated) and included that as a variable in our regression models for predicting nARMSS. In the regression models for predicting NEDA-3 during 2 years follow-up, we only included the treatment period of up to 2 years after the baseline sampling. Correlations between treatment duration index and protein expression were assessed using SCC (spearmanr from the python package SciPy). If treatment duration index was related to disease activity or disability worsening was assessed using two-sided Mann–Whitney U test (mannwhitneyu from the python package SciPy) and SCC, respectively.

## Logistic regression models

To build the logistic regression models predicting binary outcomes, i.e., pwMS *versus* HC, and NEDA *versus* EDA, we started from the 52 proteins that had shown to be differentially expressed between pwMS and HC in the discovery cohort. In addition, age of the pwMS at baseline and sex were included as possible features. Feature selection was performed using the functions glm and step from the R-package stats (v3.6.2)[74]. Forward selection, selecting features resulting in the maximum Akaike information criterion, was followed by backward selection, removing features until the coefficients of all features were significant ($p < 0.05$). The obtained predictions were compared with the actual values, using the score AUC, to assess the performance of the model. AUC and associated p-values were calculated using the function roc.area from the R-package verification (v1.42)[75].

## Prediction cut-off and accuracy for logistic regression models

The logistic regression model is utilized to predict the probability of a binary outcome for each individual observation. To classify these predictions, a cut-off value is established. The optimal cut-off, at which the model's accuracy is highest, was determined by utilizing the R package cutpointr (v1.1.2)[76]. Accuracy is calculated as the ratio of correctly classified observations (true positives and true negatives) to the total number of observations. When using a single protein as a predictor, each prediction corresponds to a specific level of that protein. Therefore, the protein level at the optimal cut-off is also reported.

## Transforming the NPX value to pg/ml

The levels of NfL in pwMS ($n = 38$) were measured using an additional proteomics assay, Simoa, and were reported in units of pg/ml[28]. The results of these measurements were found to be highly correlated with the NPX values obtained using Olink Explore (SCC = 0.97, $p = 2*10^{-16}$), suggesting that a linear regression model could be used (intercept = −7.745, coefficient = 0.965) to transform the NPX values to pg/ml.

## nARMSS definition

For each person we calculated an nARMSS score according to the procedure described by Manouchehrinia et al.[31]. nARMSS is a score which quantifies the overall disability worsening of a person, normalized to the person's age and follow-up time. pwMS with less than two EDSS scores or two or more EDSS scores over a shorter period than 3 years were excluded. First, the EDSS scores were transformed to ARMSS scores using the global ARMSS matrix ($n = 25,558$) from ref. 31. Second, the nARMSS scores were calculated according to the formula:

$$\text{nARMSS} = \frac{1}{(\text{age}_n - \text{age}_1)} \left( \int_{\text{age}_1}^{\text{age}_n} \text{ARMSS}_{\text{age}} - [(\text{age}_n - \text{age}_1) \cdot 5] \right) \quad (1)$$

The integral was calculated using the trapezoid method from the python package SciPy. The nARMSS scores are normalized to the range [−5, 5], where a score of 0 represents the average disability worsening of pwMS based on historical MS cohorts presented in the global ARMSS matrix. The nARMSS scores were correlated to the first ARMSS scores and last ARMSS scores using SCC. If there was a significant difference between the SCCs was assessed using z-test on Fisher's transformed correlation coefficients and p-values were obtained using a one-sided permutation test.

## Linear regression model for nARMSS prediction

A linear regression model to predict nARMSS from the baseline protein expression values was trained using the function LinearRegression from the python package scikit-learn (v1.1.2)[73]. Feature selection was performed in three steps:

(1) Selecting the 52 proteins that were differentially expressed in pwMS compared to HC. In addition, the age of the pwMS at baseline and sex were included as possible features.

(2) Forward selection. Features were added one at a time, according to which feature resulted in the greatest increase in $R^2$ score, using $R^2$ scores obtained from leave-one-out cross validation. Features were added until a maximum $R^2$ score was reached. $R^2$ scores were calculated using the function r2_score from the python package scikit-learn.

(3) Backward selection. Features were removed one at a time until the coefficients of all features were significant ($p < 0.05$). After removing a feature, coefficients were recalculated. Coefficients and corresponding p-values were calculated using the function OLS from the python package statsmodels (v0.13.2)[77].

The performance of the selected model was assessed using SCC (spearmanr from the python package SciPy) and CCC between the true nARMSS score and the predicted values. The significance of CCC was calculated using t-statistics. In addition, the performance of the model to predict groups of pwMS with similar nARMSS scores were assessed using AUC and accuracy. The pwMS were divided into two groups using three different thresholds: nARMSS < −4, nARMSS < −3, and

nARMSS > −1. Before being used to calculate AUC scores, the predicted nARMSS scores in the range [−5, 5] were scaled to the range [0, 1]. For the thresholds nARMSS < −4 and nARMSS < −3 we used 1 − prediction when calculating AUC scores. AUC scores were calculated using the function roc_auc_area from the python package scikit-learn. The significance of the AUC scores was assessed using one-sided Mann–Whitney U test (mannwhitneyu from the python package SciPy).

### Network analysis and enrichment analysis

The proteins were connected using STRING version 11.5[32]. We used interactions with a minimum combined interaction score of 0.4 (medium confidence, all interaction sources). One intermediate protein was allowed to connect proteins by setting the parameter 1st shell to max 10 interactors. This connected all proteins except FCN2 and ADA2. FCN2 was connected to the network with intermediate protein PTX3 (combined interaction score > 0.4). To understand the functional context of the proteins we first performed Gene Ontology enrichment analysis and a KEGG pathway enrichment analysis using the R-package clusterProfiler (v4.4.4)[78]. Significant functional terms ($p < 0.05$) that were similar in terms of their main function were put under the same generic category to better understand the functions of the proteins as a network. Proteins that could not be annotated in this way were chosen for different functional categories based on their functions described in the literature (references in Supplementary Table 6).

### Reporting summary

Further information on research design is available in the Nature Portfolio Reporting Summary linked to this article.

## Data availability

The proteomics data generated in this study have been deposited in the DiVA (Digitala Vetenskapliga Arkivet) portal under identifier https://doi.org/10.48360/jcps-gw67[79]. The proteomics data is available under restricted access due to data privacy regulations aimed at protecting sensitive personal information, access can be obtained by contacting mika.gustafsson@liu.se. Please note that access will be granted after an evaluation of accordance with Swedish legislation. We anticipate that the data will become available within 2 weeks after requested access. Publicly available datasets used in this study: MS-associated genes (C0026769) from DisGeNet version 7.0 (https://www.disgenet.org/)[24], MS SNPs from GWAS[25], global ARMSS matrix[31], and human protein–protein interactions from STRING version 11.5 (https://string-db.org/)[32]. The authors declare that all other data supporting the findings of this study are available within the paper and its supplementary information files. Source data are provided with this paper.

## Code availability

The code used for data analysis is available in Zenodo with the identifier https://doi.org/10.5281/zenodo.8370589[80].

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

## Acknowledgements

The study was funded by the Swedish Foundation for Strategic Research (SB16-0011 [M.G., J.E.]), the Swedish Brain Foundation, Knut and Alice Wallenberg Foundation, and Margareth AF Ugglas Foundation, Swedish Research Council (2019-04193 [M.G.], 2018-02776 [J.E.], 2020-02700 [F.P.], 2020-00014 [Z.L.P.], 2021-03092 [J.E.]), the Medical Research Council of Southeast Sweden (FORSS-315121 [J.E.]), NEURO Sweden (F2018-0052 [J.E.]), ALF grants, Region Östergötland, the Swedish Foundation for MS Research and the European Union's Marie Sklodowska-Curie (813863 [J.E.]). The authors would like to acknowledge support of the Clinical biomarker facility at SciLifeLab Sweden for providing assistance in protein analyses.

## Author contributions

S. Hellberg, J.R., and M.K. handled blood samples and prepared samples for proteomics profiling. J.Å. and S. Hojjati performed all analysis under supervision of M.G. and J.E.. J.M., J.E., T.O., and F.P. designed the clinical study and J.M., R.R., T.O., F.P., and M.K. recruited patients and compiled all the clinical data used in the study. J.M., F.P., T.O., J.E., and M.G. were involved in the overall design and supervised the study together with C.A., I.K., M.C.J., and Z.L.P. J.Å., S. Hojjati, S. Hellberg, J.E., and M.G. were responsible for drafting the manuscript. All authors have read and revised the article and approve the submitted version.

## Funding

## Competing interests

T.O. has received advisory board/lecture honoraria as well as unrestricted research grants from Biogen, Novartis, Sanofi, and Merck. None of which has any relation to the current manuscript. F.P. has received research grants from Janssen, Merck KgaA and UCB, and fees for serving on DMC in clinical trials with Chugai, Lundbeck and Roche, and preparation of expert witness statement for Novartis. J.M. has received honoraria for advisory boards for Sanofi Genzyme and Merck and lecture honorarium from Merck. The remaining authors declare no competing interests.

## Ethics statement

This study was reviewed and approved by the regional ethics review board in Linköping Sweden (2013/155-32, 2016/304-32, 2016/305-32, 2014/311-31, 2017/288-31) and the ethics review board in Stockholm Sweden (2022-03650-02). The participants provided their written informed consent to participate in this study.
