## [Peer Review File · Nature Communications]

REVIEWER COMMENTS

Reviewer #1 (expert in neurology, multiple sclerosis and biomarkers for multiple sclerosis):

Proteomics profiling reveals biomarkers for predicting diagnosis, disease activity and long-term disability outcome in multiple sclerosis

The authors measured CSF and plasma proteome in 2 cohorts of early CIS/RRMS patients and Healthy controls (HC) using Olink Explore panels that in total measured close to 1500 unique proteins, although about 1000 had detectable CSF levels in at least 25% of subjects.

They assembled biomarkers into several linear models:

1. Model that differentiated MS from HD
2. Model that predicted short-term disease activity measured by NEDA3 over 2 years
3. Model that predicted normalized ARMSS (nARMSS) based on longitudinal data that comprised median follow-up of 5.4 years in the larger/training cohort and 3.1 years in the smaller/replication cohort.

The main results can be summarized as:

1. CSF proteome, but not plasma proteome differentiated MS from HD
2. CSF NFL represented best "model" to predict 2-year NEDA
3. The most important result is that a linear combination of 11 CSF proteins showed significant and reasonably strong correlation with nARMSS (Spearman Rho = 0.69 in training and 0.74 in replication cohorts). Similar positive results were obtained when nARMSS was dichotomized on 3 different thresholds and the authors calculated AUCs.

The strength of the study rests in the result validation in the independent cohort. Indeed, only ~8% of publish studies use independent validation in MS field

<https://pubmed.ncbi.nlm.nih.gov/35720> . The other strength is depositing raw data and analysis code as supplementary information – this too is done rarely. Unfortunately, the deposited data are not complete – the Excel files contain biomarker data but lack age, gender, details of therapy info, ARMSS, nARMSS (although some of these can be parsed together from different tables).

Main criticism:

1. The introduction (and discussion) is inadequate – the authors fail to correctly put their results in the relationship with published literature. They state how their study is "unprecedented", but fail to mention/cite multiple studies from multiple laboratories that already obtained (and independently validated) clinical value of overlapping CSF proteins (in fact, most proteins in their final prognostic classifier) such as IL12p40, CXCL13, sCD27, MMP9, CHI3L1 or SLAMF7, or studies that already validated CSF-biomarker-based MS diagnostic test and prognostic classifiers. Even Table S2 titled "Known MS biomarkers with supporting references" omits most of these data/references. Similarly, there is a wealth of literature on NFL predicting contrast-enhancing lesions (CELs), relapses or both, from many different labs (including Karolinska) and some of these papers include independent validation cohorts and AUCs. The NEDA results from this study must be discussed in the context of those publications.
2. Differentiating MS from HD is actually not difficult for a clinician. Differentiating MS from other inflammatory neurological diseases (OIND) or non-inflammatory neurological diseases (NIND) that mimic MS is difficult. CSF-biomarker-based, independently-validated models for differentiating MS from OIND and NIND controls, and even differentiating RRMS from progressive MS has been published in 2017: <https://www.ncbi.nlm.nih.gov/pmc/articles/PMC5743213/> and should not be omitted from introduction or discussion. The accuracy of published classifier, in the independent validation cohort greatly outperformed results obtained in the current study.
3. The most surprising observation related to the most important result of current study is stronger performance of the prognostic classifier in the (small, n=33) independent validation cohort as compared to training cohort. In the vast experience of this reviewer, that virtually never happens, as machine-learning models, even those based on linear regression, tend to overfit data unless they are trained on very large cohorts. The reviewer's experience is fully supported by recent meta-analysis of the MS modeling studies: <https://pubmed.ncbi.nlm.nih.gov/35720> where all models achieved weaker independent cohort effect sizes in comparison to training cohort data.

This unexpected observation made me search carefully for potential sources of "bias":

a. Lack of blinding

b. Arbitrary decisions, possibly/likely? made after analysis of the independent validation cohort: e.g. how did the authors decide which drugs they classified as low versus high efficacy? Is their classification supported by objective data? E.g., there was a meta-analysis of effect sizes of MS drugs performed and published <https://pubmed.ncbi.nlm.nih.gov/29176956/> which found daclizumab being high efficacy and fingolimod low efficacy, while the authors used opposite classification. The reviewer is unsure if this influenced analyses or not. Similarly, how did the authors decide that for the nARMSS models they will include only MS patients with at least 2 EDSS collected over 3 years and thus excluded 18 (35%) of patients from their independent validation cohort? Could the authors show correlation between ARMSS and nARMSS for all 51 MS patients in the validation cohort? The reviewer believes that the correlation between ARMSS and nARMSS will be significant and strong for both cohorts, because most patients (based on data presented in Supplementary information) did not meaningfully progress during follow-up. There were 3 patients with severe MS who increased EDSS by many points over the follow-up, but these were exceptions. If ARMSS and nARMSS correlate strongly, then no patients should be omitted from validation of the models.

c. Problem with "contaminating validation cohort". The correct use of validation cohort is to refrain from analyzing it during "data mining" process. All "data mining", i.e., all alterations to the models should be done solely in the training cohort till final models are selected, power is calculated and models are applied to the independent validation cohort. Instead, it is frequent, and it seemingly happened here as well, that the independent validation cohort results were considered at every analysis step and actually influenced the analyses: For example, on page 10 the authors present IL12p40 as the "best model" (i.e., best single biomarker) to differentiate MS from HD. It achieved AUC=0.93 in the training and AUC=0.92 in the validation cohort. However, their selected pipeline identified TNF and MZB1 combination as best model (AUC=0.99), but this model replicated with lower AUC (0.87) than IL12p40 alone. That is circular argument: the authors must select best model based on training cohort data and then report only validation of that model. Otherwise, the reviewer (and the reader?) ends up wondering about how many other analyses were performed and rejected based on independent validation results and are neither reported here nor p-values adjusted for these additional analyses. Similar circular argument is presented at page 11 (NFL alone versus NFL, IL-18, PDCD1 and CD6 model).

d. The biggest problem with both NEDA3 and sARMSS models is the treatment issue: Although the authors state that all data related to this paper are deposited to supplementary information, I could not identify data on treatments. Table 1 shows me that 5/87 patients in the training cohort and 0/51 patients in the validation cohort started treatments before the LP, but I could find no data that would tell me which patients were treated with which medications during follow-up. The authors describe "treatment index" but this is based on some arbitrary classification of drugs into high and low efficacy as I indicated above. I think this is absolutely essential information, because we know that current DMTs inhibit development of NEDA3 or subsequent disability progression and they do latter more effectively in younger MS patients than older. There are also CSF biomarker studies published that show that levels of some of the proteins that were used in the prognostic classifier here (i.e., CXCL13, CHIT3L1, MMP9 etc) are in fact inhibited by FDA-approved DMTs. The authors state that including their therapy index did not make much difference in results (page 16), but this makes logically no sense for NEDA3 and even for nARMSS, unless their nARMSS model does not prognosticate future rates of disability progression but instead measures past rate of disability progression. This may, indeed, be the case and it should be investigated by correlating nARMSS CSF model to ARMSS measured at the time of LP, as well as investigating correlations between nARMSS and ARMSS measured at the time of LP in all patients: the reviewer expects both being significant with reasonably high effect sizes. In fact, looking at the final model (Table S4) and directionality of the coefficients, it is obvious to me that the model predicts lower nARMSS in older patients without contrast-enhancing lesions (CELs), by assigning negative directionality to Age and proteins that correlate with CELs, such as CXCL13, ICAM3, and CHI3L1. So, the nARMSS model looks backwards in time, when all patients in the validation cohort (at least 33 out of 51 included in the validation here) were untreated and, consequently, patients with active MS accumulated more disability.

e. Somewhat minor, although relevant point is that Spearman Rho is not a correct measure of model's performance; the Linh's concordance coefficient (also from 0-1 that assesses how well the model predicts 1:1 line) is the correct assessment. For example, Figure S5 shows distribution of

nARMSS for training and validation cohorts: we see lack of Gaussian distribution for both (whereas nARMSS is calculated so that unbiased cohorts that reflect underlying population should have Gaussian distribution centered around 0), but we see clearly that 3 individuals in the replication cohort have nARMSS higher than 2.5 and total of 8 MS patients have nARMSS higher than 0. Yet, when we examine model performance in Figure 5a, right panel, we see only one patient with CSF-predicted nARMSS above 0, except it is a patient who has real nARMSS 0. In other words, relatively strong Spearman Rho is misleading because it correctly classified 0/8 patients with higher-than-average measured nARMSS as having CSF-predicted nARMSS higher than population average. Instead, it incorrectly classified one patients with nARMSS aligned with population mean (i.e., nARMSS ~0) as having unusually high rates of MS progression (i.e., CSF predicted nARMSS ~4). This would be obviously problematic for a prognostic classifier that is supposed to be used at patient level.

f. Another minor issue: Plasma NFL have stronger predictive power for predicting nARMSS than CSF NFL – this again has been published before <https://pubmed.ncbi.nlm.nih.gov/35737460/>, so current study represents nice external validation.

Overall, this study independently validates importance of several CSF biomarkers in differentiating MS from HD and predicting MS activity. It also adds to the increasing evidence that clinical value of serum/plasma biomarkers in MS field is limited.

Providing prognostic information is extremely important goal of CSF biomarkers and this study also makes an important step towards that goal. However, the authors need to address several aforementioned shortcomings and put their study in the context of already-published studies on the same topic.

Bibi Bielekova

Reviewer #2 (expert in neuroscience and biomarkers):

The authors looked at whether using baseline samples and Olink/PEA-NGS to differentiate and predict short-term disease activity and long-term disease progression. The manuscript is well-written and the results are clearly presented. Nevertheless, I have several recommendations and suggestions for further improvement.

-It is unclear how the cohorts were defined. Please verify that there are no significant differences between the cohorts for the characteristics included in Table 1.

- Medication was used by the MS patients during follow-up.
 - o 5 patients were already taking medication (2x NTZ, 2x RTX and 1x DMF). Did the authors check for treatment effects on the disease activity predicting biomarkers? These samples might be excluded since their protein profile may be different from treatment-naïve patients. Same for the 9 patients who underwent steroid treatment.
 - o They lump all 1st line treatments and 2nd line treatments together. This means that drugs such as interferon end up with dimethyl fumarate and teriflunomide, and fingolimod, NTZ, OCR and aHSCT with the other. Then they used only the 2nd line treatments, for a treatment time index. This method reduces the granulation and detailed information that can be obtained from the results, since every treatment has a different effect on the biology, and disease worsening. The groups sizes should be bigger to draw meaningful results.
 - o It is not explained how many patients used which treatment. This must be mentioned to allow meaningful conclusions.
- No adjustment has been made for disease duration. The median ranges from 0-136 months. It was not clear nor justified why disease duration was not adjusted for.
- 62/92 patients had a CIS at baseline in the discovery cohort, 21/51 in the replication cohort. It was not clear whether these patients actually developed RRMS. Otherwise you are looking at CIS and not so much PwMS/RRMS. This is therefore also relevant missing information, which affect the conclusions.
- NFL is defined as predictor of disease activity with a value of 737 pg/ml. GFAP is not a significant biomarker in any of the analyses. Can the authors explain this discrepancy with other literature?
- How often was EDSS administered and at what intervals?
- Sentence 521 through 534: first sentence describes that they did not include proteins below the

LLOD in >75% of the samples. In the following sentence, they state that they have included the remaining values under the LLOD. This is a contradiction. Moreover, the removal could lead to discarding potentially valuable markers, especially because the HC group is smaller than the MS group. For example, if a protein is below LOD for all MS but no HC it would be excluded based on the 75% since the MS group covers 80% of the discovery cohort. I advise applying this 75% rule to each group separately within the discovery cohort.

-“Plasma samples from (21) pwMS in the replication cohort had higher expression of several protein markers known to be affected by sampling and handling variability (22) and were therefore excluded from further analysis” Please, include the PCA plot showing that these samples are indeed a separate cluster based on all proteins and justifying the exclusion of these samples.

-Figure1a; are PCA plots based on all proteins, or significant proteins only? I assume the first, likely, similar separation patterns will turn up after selecting the significant ($P < 0.05$) proteins only. Please also color the PCA plots based on the institute of inclusion to show no center effect is present.

-“In contrast to CSF, protein profiling of plasma did not reveal any significant differences in protein expression after FDR in pwMS compared with HC (Fig. 2a, b).” How do the 52 biomarkers identified in CSF correlate with the same markers in plasma? Because the signal in plasma is lower and the cohort is relatively small, it might be a power issue (especially since the authors excluded 21 plasma samples) that these markers are not significant in plasma. Despite the observed higher values in plasma for these 21 excluded samples, how is the overlap of the 52 proteins when not excluding these 21 plasma samples?

-The authors state that IL-12p40 is a robust marker for MS diagnosis. However, this seems a bold statement since IL-12p40 is associated with many inflammatory responses including asthma (DOI:<https://eur04.safelinks.protection.outlook.com/?url=https%3A//doi.org/10.1016/j.it.2006.11.002&data=05%7C01%7Cc.teunissen@amsterdamumc.nl%7C31a7adae5bfa410f742008db56bdb2a4%7C68dfab1a11bb4cc6beb528d756984fb6%7C0%7C0%7C638199144974004831%7CUnknown%7CTWFpbGZsb3d8eyJWIjoic4wLjAwMDAiLCJQIjoiV2luMzIiLCJBTiI6Ikk1haWwiLCJXVCI6Mn0=%7C3000%7C%7C%7C&sdata=TxuBDh4maE1gCRAS3ACOTtnjey2%2B/CNYH%2B5jxcwqcyU=&reserved=0>). Did the authors check for chronic diseases and comorbidities in the MS and control groups? To my opinion, these markers should be validated in a differential diagnosis cohort first, before such a strong claim can be written down. This could also be part of the discussion section.

-Figure 6a: it is unclear what the meaning is of the linewidth between the circles. I propose a different layout, for example, circular to get a better view of the interactions. Furthermore, it would be interesting to add a KEGG pathway enrichment analysis to relate the proteins to specific pathways. This seems more informative than related individual proteins in understanding the mechanisms of disease.

Data availability: I advise uploading the raw Olink data and annotation data to a data repository to allow replication of the findings by other scientists.

RESPONSE TO REVIEWERS' COMMENTS

Reviewer #1 (expert in neurology, multiple sclerosis and biomarkers for multiple sclerosis):

Proteomics profiling reveals biomarkers for predicting diagnosis, disease activity and long-term disability outcome in multiple sclerosis

The authors measured CSF and plasma proteome in 2 cohorts of early CIS/RRMS patients and Healthy controls (HC) using Olink Explore panels that in total measured close to 1500 unique proteins, although about 1000 had detectable CSF levels in at least 25% of subjects.

They assembled biomarkers into several linear models:

1. Model that differentiated MS from HD
2. Model that predicted short-term disease activity measured by NEDA3 over 2 years
3. Model that predicted normalized ARMSS (nARMSS) based on longitudinal data that comprised median follow-up of 5.4 years in the larger/training cohort and 3.1 years in the smaller/replication cohort.

The main results can be summarized as:

1. CSF proteome, but not plasma proteome differentiated MS from HD
2. CSF NFL represented best "model" to predict 2-year NEDA
3. The most important result is that a linear combination of 11 CSF proteins showed significant and reasonably strong correlation with nARMSS (Spearman Rho = 0.69 in training and 0.74 in replication cohorts). Similar positive results were obtained when nARMSS was dichotomized on 3 different thresholds and the authors calculated AUCs.

The strength of the study rests in the result validation in the independent cohort. Indeed, only ~8% of publish studies use independent validation in MS field

<https://pubmed.ncbi.nlm.nih.gov/35720> . The other strength is depositing raw data and analysis code as supplementary information – this too is done rarely. Unfortunately, the deposited data are not complete – the Excel files contain biomarker data but lack age, gender, details of therapy info, ARMSS, nARMSS (although some of these can be parsed together from different tables).

Response: Ensuring the reproducibility of our research findings is of utmost importance, and thus, we are dedicated to depositing comprehensive raw data in accordance with our ethical permits. Supplementary Data 2 has been thoroughly updated to include comprehensive data on the characteristics of each individual. This updated table now provides detailed information for every individual, including their sex, age, baseline diagnosis, whether individuals with CIS developed RRMS, follow-up time from baseline, disease activity status after two years (NEDA/EDA), nARRMS score, CSF albumin ratio, disease duration (time from first symptom to sampling), second line treatment index, first line treatment index, total duration of observation, second line treatment duration, first line treatment duration, specific names of all treatments administered, type of each treatment (first line or second line), duration of each specific treatment, all EDSS scores, all ARMSS scores, and intervals for EDSS assessments from baseline.

Main criticism:

1. The introduction (and discussion) is inadequate – the authors fail to correctly put their results in the relationship with published literature. They state how their study is

“unprecedented”, but fail to mention/cite multiple studies from multiple laboratories that already obtained (and independently validated) clinical value of overlapping CSF proteins (in fact, most proteins in their final prognostic classifier) such as IL12p40, CXCL13, sCD27, MMP9, CHI3L1 or SLAMF7, or studies that already validated CSF-biomarker-based MS diagnostic test and prognostic classifiers. Even Table S2 titled “Known MS biomarkers with supporting references” omits most of these data/references. Similarly, there is a wealth of literature on NFL predicting contrast-enhancing lesions (CELs), relapses or both, from many different labs (including Karolinska) and some of these papers include independent validation cohorts and AUCs. The NEDA results from this study must be discussed in the context of those publications.

Response: We are very grateful for these valuable suggestions, and we fully agree with the reviewer that the introduction and discussion part of our study lacked some key references on other CSF-based MS biomarkers and classifiers. Therefore, we have improved these sections accordingly. In particular:

1. In the introduction, we revised our statement regarding previous MS models and removed the term "unprecedented". Nonetheless, in lines 73 to 76 we referenced a meta-analysis study (Liu, J. *et al.* *Frontiers in Neurology* 2022) that highlights the scarcity of studies incorporating a separate validation cohort, with less than 8% of studies following this practice.
2. In the introduction, from line 43 to 45, we included an example of an extensive proteomics study in CSF: (Kosa, P. *et al.* *Nature Communications* 2022).
3. We found the study (Barbour, C. *et al.* *Annals of Neurology* 2017) of high relevance for our discussion section about the ability of our suggested proteins (MZB1, CD79B, CD27 and TNFRSF13B) for differentiating between MS and HCs. In lines 512 to 521, we have mentioned and cited this article because it emphasizes the significance of B cell activation, a concept that aligns with our identified proteins and since it successfully distinguishes between MS and ONDs using other proteins related to expansion and activation of B cells. We have also explicitly acknowledged the need for further investigation into whether our suggested proteins can differentiate between MS and other neurological diseases (ONDs).
4. In our discussion on NfL, we have now added a supporting study (Kosa, P. *et al.* *JCI Insight* 2022) which also showed the higher predictive power of plasma NfL compared to CSF NfL for disability in line 489 to 491. We have also included Masvekar, R. *et al.* *Frontiers in Neuroscience* 2021; Novakova *et al.* *Multiple Sclerosis and Related Disorders* 2020 and Gil-Perotin, S. *et al.* *Frontiers in Neurology* 2019, in our references for studies that showed that NfL has predictive power for contrast-enhancing lesions, relapses or both in line 479 to 480.
5. A new paragraph from line 471 to 475 has been added and a number of studies have been referenced to show the clinical value of some of the more well-established proteins in the final classifier for disability worsening. The added references (also added to Supplementary Table 1 on known MS biomarkers) are: Masvekar, R. *et al.* *Frontiers in Neuroscience* 2021; Lucchini, M. *et al.* *Molecular Neurobiology* 2022; Novakova, L. *et al.* *Multiple Sclerosis and Related Disorders* 2020; Lin, J. *et al.* *Brain* 2023; Pachner, A. *et al.* *Biomedicines* 2022.

2. Differentiating MS from HD is actually not difficult for a clinician. Differentiating MS from other inflammatory neurological diseases (OIND) or non-inflammatory neurological diseases

(NIND) that mimic MS is difficult. CSF-biomarker-based, independently-validated models for differentiating MS from OIND and NIND controls, and even differentiating RRMS from progressive MS has been published in 2017: <https://www.ncbi.nlm.nih.gov/pmc/articles/PMC5743213/> and should not be omitted from introduction or discussion. The accuracy of published classifier, in the independent validation cohort greatly outperformed results obtained in the current study.

Response: We found the reference (Barbour, C. *et al.* Annals of Neurology 2017) suggested by the reviewer informative and we have added this to the discussion part of the manuscript in lines 512 to 521. In addition, looking at the most predictive proteins, namely MZB1, CD27, CD79B and TNFRSF13B in our paper, which are all associated with B cell activation, we believe that our findings align with the conclusion presented in the referenced article, albeit with different proteins. The statement *"The essential difference between MS and its mimics is selective expansion/activation of B cell/plasma-cell lineages, out of proportion to the activation of other immune cells and to the resultant injury/stress of CNS-resident cells"* supports the notion that the B cell activation markers identified in our study as biomarkers for discriminating between MS and controls could potentially demonstrate significance in distinguishing between MS and other neurological disorders in future studies. However, we acknowledge that this is merely a possibility. To clarify our stance on this matter, we have also included *"Whether the proteins identified in our study will prove useful for differentiating MS from ONDs remains to be settled."* in line 520 to 521.

3. The most surprising observation related to the most important result of current study is stronger performance of the prognostic classifier in the (small, n=33) independent validation cohort as compared to training cohort. In the vast experience of this reviewer, that virtually never happens, as machine-learning models, even those based on linear regression, tend to overfit data unless they are trained on very large cohorts. The reviewer's experience is fully supported by recent meta-analysis of the MS modeling studies: <https://pubmed.ncbi.nlm.nih.gov/35720> where all models achieved weaker independent cohort effect sizes in comparison to training cohort data. This unexpected observation made me search carefully for potential sources of "bias":

Response: As the reviewer points out, it is uncommon that machine-learning models perform better in a validation cohort than the training cohort, but it is not impossible. We have carefully considered the reviewer's concerns regarding possible "bias" and have addressed each concern point by point below. However, we have performed a z-test on the Fisher's transformed correlations coefficients for the correlations between the true nARMSS and predicted nARMSSS for the discovery cohort (SCC = 0.69) and replication cohort (SCC = 0.74) and found that the differences between the two are non-significant ($p = 0.32$).

a. Lack of blinding

Response: We are very grateful for these valuable suggestions, and we agree with the reviewer's emphasis on the importance of blinding, particularly from an analytical standpoint. In our study design, we took great care to mitigate bias by strictly separating the discovery and replication cohorts. All models were trained exclusively on the discovery cohort and subsequently validated on the replication cohort, ensuring no cross-talk or re-adjustment based on the replication cohort data. This strategy facilitated an effective form of blinding at

the model level. We acknowledge that our initial presentation may have led to some confusion regarding this process, as we concurrently presented the best univariate markers for both the discovery and replication cohorts. While this was done to highlight clinically interesting points, it may have obscured the strict division between the cohorts. To rectify this, we have now revised the biomarker figures (Figure 3) and accompanying text to clearly delineate the independence of the two cohorts, thereby underscoring that no data leakage occurred. We believe these changes will provide a more transparent and accurate depiction of our methodology.

b. Arbitrary decisions, possibly/likely? made after analysis of the independent validation cohort: e.g. how did the authors decide which drugs they classified as low versus high efficacy? Is their classification supported by objective data? E.g., there was a meta-analysis of effect sizes of MS drugs performed and published <https://pubmed.ncbi.nlm.nih.gov/29176956/> which found daclizumab being high efficacy and fingolimod low efficacy, while the authors used opposite classification. The reviewer is unsure if this influenced analyses or not. Similarly, how did the authors decide that for the nARMSS models they will include only MS patients with at least 2 EDSS collected over 3 years and thus excluded 18 (35%) of patients from their independent validation cohort? Could the authors show correlation between ARMSS and nARMSS for all 51 MS patients in the validation cohort? The reviewer believes that the correlation between ARMSS and nARMSS will be significant and strong for both cohorts, because most patients (based on data presented in Supplementary information) did not meaningfully progress during follow-up. There were 3 patients with severe MS who increased EDSS by many points over the follow-up, but these were exceptions. If ARMSS and nARMSS correlate strongly, then no patients should be omitted from validation of the models.

Response: The classification of drugs as low- and high-efficacy was based on literature (references have been added in Supplementary Table 6) together with clinicians experienced with MS treatments. In the supplementary table we have also included a link to a document from the Swedish MS Association (unfortunately only available in Swedish). On pages 3 and 4, the treatment classifications are listed as low- (lägre effekt), medium- (måttlig effekt) or high- (hög effekt) efficacy. We included the low effect drugs as first-line treatment and the high effect drugs as second-line treatment in our study. We classified the medium effect drugs as either first-line or second-line based on the literature. Besides that, the classification of daclizumab as a low-efficacy drug in the method section of the manuscript was an error. During analysis, daclizumab was classified as a high-efficacy drug and this has now been corrected in the text (line 669). Moreover, we have seen that the literature shows discrepancy when it comes to the classification of low- and high-efficacy drugs. This is the case for Fingolimod. In this case, we based our classification on recent publications in top-ranked journals (Rotstein, D. *et al.* Nature Reviews Neurology 2019) and (Spelman, T. *et al.* JAMA Neurology 2021) in which it is classified as high-efficacy. References for the classification of all the drugs are added as Supplementary Table 6.

The decision to only include samples with at least two EDSS over three years, when calculating nARMSS, was a tradeoff between obtaining an as accurate model on the discovery cohort as possible while not losing too many patients in the replication cohort. Firstly, at least two EDSS scores are mathematically necessary for calculating an nARMSS score. In the article (Manouchehrinia, A. *et al.* Multiple Sclerosis Journal 2021), where the nARMSS score was presented, it was recommended to have a follow-up period of at least

two years, but they had also observed that the nARMSS score became more accurate with longer follow-up. We did model selection on the discovery cohort using a cutoff of two, three, and four years, which resulted in a discovery SCC of 0.63 (Lin's concordance correlation coefficient (CCC) = 0.63), 0.69 (CCC = 0.72), and 0.73 (CCC = 0.79), respectively. This suggested that we ideally should use a cutoff of four years, alternatively three years and that two years performed worst in the discovery. However, in the replication cohort the number of patients using a two year cutoff is 43, while using three years we have 33 patients, and using four years we only have 22 patients left. We decided on 3 years since the performance on the discovery cohort is good, but we can still evaluate the model on 33 patients in the replication cohort.

Regarding the reviewer's suggestion to correlate nARMSS and ARMSS, it is not possible to do it for all the 51 patients in the replication cohort, since not all of them can obtain a nARMSS score. However, we correlated the first ARMSS of each patient with nARMSS using a two-year cutoff (Discovery, n = 76; Replication, n = 43) and nARMSS using a three-year cutoff (Discovery, n = 71; Replication, n = 33). Importantly, we found a stronger correlation between nARMSS and the first ARMSS using the two-year cutoff compared to the three-year cutoff, with SCC decreasing from 0.74 to 0.71 in the discovery cohort and from 0.84 to 0.79 in the replication cohort. This indicates that a shorter follow-up period will result, not surprisingly, in a nARMSS more closely related to the first ARMSS. Therefore, we believe that it is important to use patients with a sufficient follow-up period, to obtain a score reflecting a patient's full disability progression.

c. Problem with "contaminating validation cohort". The correct use of validation cohort is to refrain from analyzing it during "data mining" process. All "data mining", i.e., all alterations to the models should be done solely in the training cohort till final models are selected, power is calculated and models are applied to the independent validation cohort. Instead, it is frequent, and it seemingly happened here as well, that the independent validation cohort results were considered at every analysis step and actually influenced the analyses: For example, on page 10 the authors present IL12p40 as the "best model" (i.e., best single biomarker) to differentiate MS from HD. It achieved AUC=0.93 in the training and AUC=0.92 in the validation cohort. However, their selected pipeline identified TNF and MZB1 combination as best model (AUC=0.99), but this model replicated with lower AUC (0.87) than IL12p40 alone. That is circular argument: the authors must select best model based on training cohort data and then report only validation of that model. Otherwise, the reviewer (and the reader?) ends up wondering about how many other analyses were performed and rejected based on independent validation results and are neither reported here nor p-values adjusted for these additional analyses. Similar circular argument is presented at page 11 (NFL alone versus NFL, IL-18, PDCD1 and CD6 model).

Response: During the model selection we were careful not to use the replication cohort when performing feature selection for the logistic regression and linear regression models (the stepwise models). However, we also believe it to be of high clinical relevance to also evaluate the proteins as individual markers, and therefore decided to also include results of the individual markers in both the discovery and the replication cohort. Many proteins achieved high performance for predicting diagnosis, and we decided to highlight the ones performing the best on both cohorts. We agree with the reviewer that this is not fully proper use of the replication cohort and have now changed the results section (line 178 to 181 and 183 to 188) and Figure 3a to highlight the stepwise model and the top proteins with the

highest performance on the discovery cohort for predicting diagnosis. The top performing proteins are MZB1, CD79B, CD27, TNFRSF13B, and IL-12p40 ordered by performance in the discovery cohort (the previously highlighted proteins were IL-12p40, TNFRSF13B, IL-12p70, and MZB1). For consistency we have also changed Figure 3b to show the stepwise model and the top proteins with the highest performance in the discovery cohort for predicting NEDA. Previously, we had excluded the proteins FASLG and CD6 since these did not perform well on the replication cohort. These proteins are now included and the order of presenting AUCs is from highest to lowest in the discovery cohort. However, the conclusion that NfL is superior for predicting disease activity remains unchanged.

d. The biggest problem with both NEDA3 and sARMSS models is the treatment issue: Although the authors state that all data related to this paper are deposited to supplementary information, I could not identify data on treatments. Table 1 shows me that 5/87 patients in the training cohort and 0/51 patients in the validation cohort started treatments before the LP, but I could find no data that would tell me which patients were treated with which medications during follow-up. The authors describe "treatment index" but this is based on some arbitrary classification of drugs into high and low efficacy as I indicated above. I think this is absolutely essential information, because we know that current DMTs inhibit development of NEDA3 or subsequent disability progression and they do latter more effectively in younger MS patients than older. There are also CSF biomarker studies published that show that levels of some of the proteins that were used in the prognostic classifier here (i.e., CXCL13, CHIT3L1, MMP9 etc) are in fact inhibited by FDA-approved DMTs. The authors state that including their therapy index did not make much difference in results (page 16), but this makes logically no sense for NEDA3 and even for nARMSS, unless their nARMSS model does not prognosticate future rates of disability progression but instead measures past rate of disability progression. This may, indeed, be the case and it should be investigated by correlating nARMSS CSF model to ARMSS measured at the time of LP, as well as investigating correlations between nARMSS and ARMSS measured at the time of LP in all patients: the reviewer expects both being significant with reasonably high effect sizes. In fact, looking at the final model (Table S4) and directionality of the coefficients, it is obvious to me that the model predicts lower nARMSS in older patients without contrast-enhancing lesions (CELs), by assigning negative directionality to Age and proteins that correlate with CELs, such as CXCL13, ICAM3, and CHI3L1. So, the nARMSS model looks backwards in time, when all patients in the validation cohort (at least 33 out of 51 included in the validation here) were untreated and, consequently, patients with active MS accumulated more disability.

Response: The potential problem with the influence of treatment in the models is an important issue raised by the reviewer. It is not feasible to perform a study where treatment is not an influencing factor, since persons with MS need to be treated for their symptoms. The treatments will also be different for different persons and vary during the disease course. Hence, treatment is a very complex covariate to consider. Moreover, the observational nature of our study means that we cannot draw any conclusions about the causal effect of treatment on any of the outcomes. This has been discussed in a paragraph which is now added to the discussion section of the manuscript (line 492 to 504). In general, including a feature in a machine learning model that is constructed based on the outcome variable can introduce a form of data leakage, known as "target leakage". In our study, treatment was given/changed, as expected, based on the patients' real-time progression. As

a result, we found that patients with a more severe disease course (higher nARMSS or having evidence of disease activity) had in general longer periods of treatment with second-line treatments, *i.e.*, a higher treatment duration index (as was previously mentioned in lines 293 to 296 for nARMSS and has been added to line 232 to 235 for the disease activity in the results section). Therefore, even if treatment had been selected as a feature in our model, it would have been included solely for explanatory purposes rather than for predictive purposes. However, it is likely that persons with a severe disease course would have had an even more severe disease course had they not been on treatment, resulting in treatment dampening the effect of the identified proteins, and that the suggested models could be more accurate if treatment were not an influencing factor. In addition, we found the treatment duration index to be significantly positively correlating with the expression of many of the differentially expressed proteins in the discovery cohort (added on line 239 to 244 in the results section and as Supplementary Figure 7), which could be a reason why treatment duration index did not have a great effect on the performance of our models. It is important to note that the distributions of treatment duration index in our two cohorts are significantly different (median in discovery cohort = 0.47 vs. median in replication cohort = 0.97, $p = 7 \cdot 10^{-8}$) and none of the protein markers selected in our models for predicting disease progression showed a significant correlation with the treatment duration index in both the discovery and the replication cohort. This is not surprising since the treatments were not administered based on protein concentrations but is reassuring that the model is not predicting the future treatment which would be off target. The fact that despite the difference in treatment portfolio between the two cohorts the model is still showing good performance in the replication cohort, shows that the proteins have stronger predictive power than the treatment duration index for disease progression.

We believe that nARMSS both measure future and past rates of disability. To address the concern raised by the reviewer, we investigated the correlations of nARMSS with the first ARMSS (close to baseline sampling) and the last ARMSS used for calculating the nARMSS scores. Already with the first ARMSS we observe a strong correlation with nARMSS, which is part of how the score is designed. nARMSS does not only consider the rate of change in disability, but also the quantitative measure of disability, *i.e.*, a constant score of 5 will result in a higher nARMSS than an increase from 0 to 5 during the disease course. Importantly, the correlation with the last ARMSS is significantly stronger than with the first ARMSS for both the discovery cohort (SCC = 0.89 compared to SCC = 0.71, $p = 0.003$) and the replication cohort (SCC = 0.92 compared to SCC = 0.79, $p = 0.03$), indicating a more stable score with strong relation to future disability. These results are added on line 286 to 289 in the results section of the manuscript.

In conclusion, the treatment issue is a complex problem, and our study is not designed to fully capture the effect of treatment. However, the identified protein biomarkers we believe to be able to capture a patient's disease trajectory, although the full effect of the biomarkers could possibly have been dampened by treatment. We have added a more extensive discussion of treatment in the manuscript, to address the treatment issue more fully (line 492 to 504). In addition, we have added more extensive information about type of treatments and duration of treatments for each person with MS to Supplementary Data 2.

e. Somewhat minor, although relevant point is that Spearman Rho is not a correct measure of model's performance; the Linh's concordance coefficient (also from 0-1 that assesses how well the model predicts 1:1 line) is the correct assessment. For example, Figure S5 shows distribution of nARMSS for training and validation cohorts: we see lack of Gaussian

distribution for both (whereas nARMSS is calculated so that unbiased cohorts that reflect underlying population should have Gaussian distribution centered around 0), but we see clearly that 3 individuals in the replication cohort have nARMSS higher than 2.5 and total of 8 MS patients have nARMSS higher than 0. Yet, when we examine model performance in Figure 5a, right panel, we see only one patient with CSF-predicted nARMSS above 0, except it is a patient who has real nARMSS 0. In other words, relatively strong Spearman Rho is misleading because it correctly classified 0/8 patients with higher-than-average measured nARMSS as having CSF-predicted nARMSS higher than population average. Instead, it incorrectly classified one patients with nARMSS aligned with population mean (i.e., nARMSS ~0) as having unusually high rates of MS progression (i.e., CSF predicted nARMSS ~4). This would be obviously problematic for a prognostic classifier that is supposed to be used at patient level.

Response: We agree that Lin's concordance correlation coefficient (CCC) is an appropriate measure of performance of our nARMSS models and have added it as a performance metric in the manuscript. The CSF nARMSS model had a CCC of 0.72 ($p = 2 \times 10^{-12}$) in the discovery cohort and a CCC of 0.51 ($p = 0.002$) in the replication cohort. However, we do believe Spearman's correlation coefficient (SCC) to also be a valuable measure of performance. A model does not necessarily need to make accurate predictions to be informative and relevant, but the model needs to capture tendencies in the data, which is captured using SCC. Deviation from a 1:1 line can be adjusted for, as long as the predictions correspond to the same order as the true scores. Therefore, we believe that both SCC and CCC give important information as performance metrics. We have changed the results section to include CCC as a performance metric for both the CSF model (line 325 to 328) and plasma model (line 357 to 359), as well as changed the methods section (line 745 to 746), Figure 5a, 5c and Supplementary Table 4 to include CCC.

f. Another minor issue: Plasma NFL have stronger predictive power for predicting nARMSS than CSF NFL – this again has been published before <https://pubmed.ncbi.nlm.nih.gov/35737460/> , so current study represents nice external validation.

Response: We thank the reviewer for the suggested reference, which indeed is coinciding very well with what we found in our study. The reference has been added to the discussion section of our manuscript, in the sentence "*NfL in plasma showing stronger predictive power than CSF-NfL is surprising but is corroborating a recent finding that serum NfL has stronger correlation with MS severity outcomes than CSF-NfL (Kosa, P. et al. JCI Insight 2022)*" on line 489 to 491.

Overall, this study independently validates importance of several CSF biomarkers in differentiating MS from HD and predicting MS activity. It also adds to the increasing evidence that clinical value of serum/plasma biomarkers in MS field is limited. Providing prognostic information is extremely important goal of CSF biomarkers and this study also makes an important step towards that goal. However, the authors need to address several afore-mentioned shortcomings and put their study in the context of already-published studies on the same topic.

Response: We appreciate the reviewer's valuable comments on our study and recognition of its importance. Regarding the mentioned shortcomings, we have thoroughly addressed them in the revised version of the manuscript. We have provided comprehensive data on the clinical characteristics of all the individuals in our study and provided further clarification on the limitations and potential implications of our findings, including the complexities associated with treatment influence. We have included additional references in both the introduction and discussion sections of the manuscript. These references help to contextualize our study within the existing body of research on CSF biomarkers in MS and related prognostic information.

Reviewer #2 (expert in neuroscience and biomarkers):

The authors looked at whether using baseline samples and Olink/PEA-NGS to differentiate and predict short-term disease activity and long-term disease progression. The manuscript is well-written and the results are clearly presented. Nevertheless, I have several recommendations and suggestions for further improvement.

- It is unclear how the cohorts were defined. Please verify that there are no significant differences between the cohorts for the characteristics included in Table 1.

Response: According to the reviewer's request, we have verified if there are any significant differences between the two cohorts for the characteristics included in Table 1, using two-sided Fisher's exact test for contingency tables or two-sided Mann-Whitney U test for continuous values. The p-values for the comparisons between persons with MS in the two cohorts are now added to the column p-value in Table 1. There are significantly more persons diagnosed with RRMS compared to CIS in the replication cohort than in the discovery cohort ($p = 0.003$). In addition, the replication cohort have in general more disease activity manifested as a larger number of T2 lesions at baseline MRI (10-20 lesions: $p = 8 \times 10^{-4}$; >20 lesions: $p = 0.04$), higher EDSS at baseline (median 1.5 compared to 1.0, $p = 0.002$), and a lower proportion of persons with no evidence of disease activity (NEDA; $p = 5 \times 10^{-4}$). Lastly, the distributions of the treatment duration index are significantly different between the discovery cohort and the replication cohort (median 0.47 compared to 0.97, $p = 7 \times 10^{-8}$). For the remaining characteristics, we did not identify any significant difference ($p > 0.05$) between the pwMS in the two cohorts.

- Medication was used by the MS patients during follow-up.
 - 5 patients were already taking medication (2x NTZ, 2x RTX and 1x DMF). Did the authors check for treatment effects on the disease activity predicting biomarkers? These samples might be excluded since their protein profile may be different from treatment-naïve patients. Same for the 9 patients who underwent steroid treatment.
 - They lump all 1st line treatments and 2nd line treatments together. This means that drugs such as interferon end up with dimethyl fumarate and teriflunomide, and fingolimod, NTZ, OCR and aHSCT with the other. Then they used only the 2nd line treatments, for a treatment time index. This method reduces the granulation and detailed information that can be obtained from the results, since every treatment has a different effect on the biology,

and disease worsening. The groups sizes should be bigger to draw meaningful results.

- It is not explained how many patients used which treatment. This must be mentioned to allow meaningful conclusions.

Response: We checked if the patients with treatment before baseline (n = 5) had significantly different protein expression than patients without treatment but found no differentially expressed proteins (line 101 to 106 in the results section of the manuscript). Therefore, we decided to include all patients for further analysis. Similarly, patients with steroid treatment before baseline (n = 9) were also included since no differentially expressed proteins were found. In addition, we decided to include treatment before baseline as a covariate in the differential expression analysis of MS compared to HC, but it did not affect the 52 differentially expressed proteins in the discovery cohort.

Including treatment during follow-up is a very complex problem, since patients are on different types of treatment and for different periods of time. In an attempt to account for the treatment effect in our analysis, we used the treatment duration index. We acknowledge that this is a simplification of the complex effect of treatments. Moreover, the observational nature of our study means that we cannot draw any conclusions about the causal effect of treatment on any of the outcomes. In addition, we found patients with a more severe disease progression (higher nARMSS or having evidence of disease activity) to have higher treatment duration index. However, we have seen that the treatment duration index is significantly positively correlated with the expression of most of the differentially expressed proteins in the discovery cohort but not with the expression of the same proteins in the replication cohort. Therefore, we believe the identified proteins to be markers of disease progression, even though treatments could possibly have dampened the effect of these markers. A more extensive discussion regarding the treatment effect on our suggested protein biomarkers has been added to the discussion section of the manuscript (line 492 to 504).

We have added detailed information about type of treatment and time on treatment for each person with MS to Supplementary Data 2.

- No adjustment has been made for disease duration. The median ranges from 0-136 months. It was not clear nor justified why disease duration was not adjusted for.

Response: No adjustment was made for disease duration at baseline sampling. We agree with the point raised by the reviewer that this could be an important factor to consider in the analysis. We have therefore investigated if any proteins were differentially expressed based on disease duration but found no differentially expressed proteins. These results are added to the results section of the manuscript (line 101 to 104). In addition, to more thoroughly rule out a significant effect of disease duration, we performed differential expression analysis for MS compared to HC using disease duration as a covariate in the Limma model, but it resulted in the same 52 differentially expressed proteins. Lastly, we concluded that disease duration at baseline sampling had no relation to a patient's nARMSS score (SCC = 0.04) or disease activity two years after baseline sampling (p = 0.92, two-sided Mann-Whitney U test).

- 62/92 patients had a CIS at baseline in the discovery cohort, 21/51 in the replication cohort. It was not clear whether these patients actually developed RRMS. Otherwise

you are looking at CIS and not so much PwMS/RRMS. This is therefore also relevant missing information, which affect the conclusions.

Response: Supplementary Data 2 has undergone an extensive update, incorporating a wealth of additional information. Specifically, we have included details regarding each individual's baseline diagnosis (CIS or RRMS) and the subsequent development of RRMS for individuals initially diagnosed with CIS.

In the discovery cohort, consisting of 62 individuals with CIS at baseline, 51 individuals progressed to develop RRMS during the observation time. In the replication cohort, all CIS individuals ultimately transitioned to RRMS. However, one patient in the CIS diagnosis group lacked available data on RRMS diagnosis, this is designated as "NA" to denote this missing information for this individual in the table.

- NfL is defined as predictor of disease activity with a value of 737 pg/ml. GFAP is not a significant biomarker in any of the analyses. Can the authors explain this discrepancy with other literature?

Response: We did not find any significant difference in expression of GFAP between persons with MS and HC (line 121 to 124 in the results section of the manuscript). We have added Supplementary Figure 4 which shows the difference in expression in CSF samples from persons with MS and HC for all proteins in our list of known MS biomarkers (including GFAP). One possible reason could be that GFAP is more related to progressive MS (Ayrygnac, X. *et al.* Scientific Reports 2020; Barro, C. *et al.* Neurology-Neuroimmunology Neuroinflammation 2023). We have measured the protein expression in samples from persons in the early stages of MS, and some known MS biomarkers might not be applicable as early biomarkers of MS but be more related to later more progressive stages of the disease.

- How often was EDSS administered and at what intervals?

Response: We have added more extensive information about EDSS scores and at which time points the EDSS scores were obtained to Supplementary Data 2.

- Sentence 521 through 534: first sentence describes that they did not include proteins below the LLOD in >75% of the samples. In the following sentence, they state that they have included the remaining values under the LLOD. This is a contradiction. Moreover, the removal could lead to discarding potentially valuable markers, especially because the HC group is smaller than the MS group. For example, if a protein is below LOD for all MS but no HC it would be excluded based on the 75% since the MS group covers 80% of the discovery cohort. I advise applying this 75% rule to each group separately within the discovery cohort.

Response: We have clarified the sentences (line 619 to 624) in the methods section explaining the removal of proteins below LOD to, hopefully, more clearly describe the process. The second sentence now states "*For the remaining proteins with NPX values below the limit of detection in some samples, the reported NPX values were kept in the data unchanged.*". Furthermore, it is true that unbalanced groups could lead to proteins below LOD being discarded based on only the MS samples. According to the reviewer's

suggestion, we have investigated what happens if we consider the groups separately when removing proteins below LOD. If we consider the percentage of samples below LOD for the 37 proteins that could potentially be added, we do not find any great imbalance between the groups. These results have been added as Supplementary Figure 12. For example, IL1B has one of the greatest imbalances with 80% of the MS samples below LOD and 61% of the HC samples below LOD in the discovery cohort. We also performed a differential expression analysis, which would have added one protein, GRAP2, in addition to the 52 proteins mentioned in the manuscript. GRAP2 is below LOD in 65% of the MS samples and 87% of the HC samples and can therefore not be considered a reliable marker. As a comparison, we also added Supplementary Figure 10 showing the portion of samples below LOD in each group for the 52 differentially expressed proteins. The proteins selected in our models are reliably expressed across the groups with for example NfL, CHI3L1, ICAM3, and TNFRSF18 above LOD in all samples and the marker SLAMF7 having the most samples below LOD with 39% of the MS samples and 74% of the HC samples below LOD in the discovery cohort. Therefore, we decided to not change the pre-processing step as it would not result in any reliable protein markers being added.

- “Plasma samples from (21) pwMS in the replication cohort had higher expression of several protein markers known to be affected by sampling and handling variability (22) and were therefore excluded from further analysis” Please, include the PCA plot showing that these samples are indeed a separate cluster based on all proteins and justifying the exclusion of these samples.

Response: According to the reviewer’s suggestion, we have added a PCA of the plasma samples from persons with MS as Supplementary Figure 2. In the PCA we can see that the majority of the plasma samples from the older cohort (group A) in the replication cohort are forming a separate cluster as expected, since it is in agreement with what the markers for sampling and handling variability also suggested as shown in Supplementary Figure 1.

- Figure 1a; are PCA plots based on all proteins, or significant proteins only? I assume the first, likely, similar separation patterns will turn up after selecting the significant ($P < 0.05$) proteins only. Please also color the PCA plots based on the institute of inclusion to show no center effect is present.

Response: In Figure 1a, the PCA plots are based on all proteins. We agree with the reviewer that this could be made more clear in the manuscript, and this has been clarified in the figure legend for Figure 2a. We also decided to change the marker shape of the samples from the replication cohort to more clearly depict the difference between the discovery cohort and the replication cohort in Figure 2a.

- “In contrast to CSF, protein profiling of plasma did not reveal any significant differences in protein expression after FDR in pwMS compared with HC (Fig. 2a, b).” How do the 52 biomarkers identified in CSF correlate with the same markers in plasma? Because the signal in plasma is lower and the cohort is relatively small, it might be a power issue (especially since the authors excluded 21 plasma samples) that these markers are not significant in plasma. Despite the observed higher values in plasma for these 21 excluded samples, how is the overlap of the 52 proteins when not excluding these 21 plasma samples?

Response: We found very poor correlations between CSF samples and plasma samples of the differentially expressed proteins in CSF. The most correlating proteins were NfL (SCC = 0.46 in the discovery cohort and SCC = 0.37 in the replication cohort) and IL18 (SCC = 0.33 in the discovery cohort and SCC = 0.41 in the replication cohort). An additional 10 proteins had correlation in the range 0.19 - 0.29 ($p < 0.05$) in the discovery cohort, whereas three also had significant correlations in the replication cohort. These results are added on line 128 to 131 in the results section of the manuscript and as Supplementary Figure 6. According to the reviewer's suggestion, we performed a differential expression analysis including the 21 plasma samples previously excluded from the replication cohort. This resulted in a much stronger signal with 26 proteins with FDR < 0.05 and 325 proteins nominally differentially expressed. However, only seven of the nominally differentially expressed proteins (one with FDR < 0.05) overlapped with the differentially expressed proteins in CSF, compared to the previous overlap of six proteins with the 21 plasma samples excluded. We believe that the 26 proteins passing FDR, should not be trusted as caused by MS, but are more likely a result of technical variability.

- The authors state that IL-12p40 is a robust marker for MS diagnosis. However, this seems a bold statement since IL-12p40 is associated with many inflammatory responses including asthma (DOI: [**Response:** We acknowledge that claiming IL-12p40 to be a robust marker for MS diagnosis was an overstatement. In our study, the focus was to predict disease progression, and not MS diagnosis. To address this issue, we have now removed the claim that we have found robust markers for MS diagnosis and changed the title of that section accordingly \(line 170 to 171 and line 178 to 188 in the results section of the manuscript\). In addition, we have expanded our discussion to more discuss the need of a comparison to differential diagnosis cohorts for such a claim \(line 512 to 521\).](https://eur04.safelinks.protection.outlook.com/?url=https%3A//doi.org/10.1016/j.it.2006.11.002&data=05%7C01%7Cc.teunissen@amsterdamumc.nl%7C31a7adae5bfa410f742008db56bdb2a4%7C68dfab1a11bb4cc6beb528d756984fb6%7C0%7C0%7C638199144974004831%7CUnknown%7CTWFpbGZsb3d8eyJWljojMC4wLjAwMDAiLCJQIjoiV2luMzliLCJBTil6Ikk1haWwiLCJXVCi6Mn0=%7C3000%7C%7C%7C&sd ata=TxuBDh4maE1gCRAS3ACOTtnjey2%2B/CNYH%2B5jxcwqcyU=&reserved=0). Did the authors check for chronic diseases and comorbidities in the MS and control groups? To my opinion, these markers should be validated in a differential diagnosis cohort first, before such a strong claim can be written down. This could also be part of the discussion section.<div data-bbox=)

- Figure 6a: it is unclear what the meaning is of the linewidth between the circles. I propose a different layout, for example, circular to get a better view of the interactions. Furthermore, it would be interesting to add a KEGG pathway enrichment analysis to relate the proteins to specific pathways. This seems more informative than related individual proteins in understanding the mechanisms of disease.

Response: The linewidth of the interactions in Figure 6a is related to the combined interaction score of the interactions in STRINGdb, which has now been clarified in the figure legend of Figure 6a. We agree that a circular layout of the network can be an informative

way to show the interactions, although not superior to the network layout in Figure 6a. We decided to not change the layout of the network in Figure 6a, but to also depict the network with circular layout in Supplementary Figure 11a, since both layouts are informative. According to the reviewer's suggestion we also added the results from a KEGG pathway enrichment analysis on line 401 to 403 in the results section of the manuscript and as supplementary figure 11b.

- Data availability: I advise uploading the raw Olink data and annotation data to a data repository to allow replication of the findings by other scientists.

Response: We fully agree with the reviewer that uploading raw data to a data repository is important for replication of research findings. We are using a new type of proteomics data (PEA-NGS), and unfortunately did not find any appropriate discipline-specific repository. We have therefore decided to deposit the raw proteomics data (Supplementary Data 3 and 4) in the repository figshare.

REVIEWER COMMENTS

Reviewer #1 (expert in neurology, multiple sclerosis and biomarkers for multiple sclerosis):
Absent.

Reviewer #2 (expert in neuroscience and biomarkers):

The authors have adequately addressed most of the issues raised, thereby significantly improving the manuscript. However, I still have one concern regarding the LOD filtering process. The authors have shown in supplemental figure 12 that the LOD filtering on the entire cohort containing imbalanced groups does not exclude potential useful biomarkers. However, as shown in supplementary figure 10, it results in a subset of biomarkers that are measured above the LOD for only a very small proportion of the samples. The worst case is for CLEC6A, of which only +/-10% of the controls have a value above the LOD and only 30% of the MS patients have a value above the LOD. Besides CLEC6A, there are 5-15 other markers with a relatively high percentage of samples with measurements below LOD. The translational ability of these markers is doubtful, both from a technical and a cost-effective perspective, and therefore have an increased chance of appearing as false positive biomarker candidates for single sample classification. Of note, NfL has no measurements below the LOD as shown in supplemental figure 10, and is a well-performing biomarker. For this reason, I advised performing the LOD filtering on each group separately instead of the entire cohort. How many of the 52 proteins have at least 75% of the samples within at least one group (either MS or control or both) above the LOD and what is the performance of this potentially translatable subpanel?

Reviewer #3 (expert in neurology and biomarkers for neurological diseases - this reviewer was asked to review the authors' responses to Reviewer #1):

The authors have responded dilligently and convincingly to the reviewer's comments and have adapted their manuscript where necessary. This has further improved the quality of the manuscript that is certainly documenting in a balanced way an important step forward in the difficult field of proteomics as a tool for classification and prognostication of MS.

RESPONSE TO REVIEWERS' COMMENTS

Reviewer #2 (expert in neuroscience and biomarkers):

The authors have adequately addressed most of the issues raised, thereby significantly improving the manuscript. However, I still have one concern regarding the LOD filtering process.

The authors have shown in supplemental figure 12 that the LOD filtering on the entire cohort containing imbalanced groups does not exclude potential useful biomarkers. However, as shown in supplementary figure 10, it results in a subset of biomarkers that are measured above the LOD for only a very small proportion of the samples. The worst case is for CLEC6A, of which only +/-10% of the controls have a value above the LOD and only 30% of the MS patients have a value above the LOD. Besides CLEC6A, there are 5-15 other markers with a relatively high percentage of samples with measurements below LOD. The translational ability of these markers is doubtful, both from a technical and a cost-effective perspective, and therefore have an increased chance of appearing as false positive biomarker candidates for single sample classification. Of note, NfL has no measurements below the LOD as shown in supplemental figure 10, and is a well-performing biomarker. For this reason, I advised performing the LOD filtering on each group separately instead of the entire cohort. How many of the 52 proteins have at least 75% of the samples within at least one group (either MS or control or both) above the LOD and what is the performance of this potentially translatable subpanel?

Response:

We are delighted that the reviewers have recognized the enhancements made to the manuscript and express our gratitude for the insightful question raised by reviewer 2. The reviewer's concern regarding the LOD filtering process is indeed valid, and we would like to provide some clarification on this matter. Firstly, it is worth noting that the manufacturer of PEA-NGS, used for the proteomics profiling, suggests that suitable exclusion limits may fall within the range of less than 25% to 50% of samples above the LOD (link to the manufacturer website). We agree with the reviewer that a high percentage of samples above the LOD is important for biomarker candidates, but a cutoff of 75% could be overly conservative and might remove important biomarker candidates. On the other hand, as the reviewer points out, those markers are more likely translatable biomarkers useful for the clinic.

Regarding the reviewer's suggestion to perform LOD filtering on each group separately, we appreciate the idea and have explored this approach. Of the 52 differentially expressed proteins 48 proteins were expressed above the LOD in more than 50% of the samples from either people with MS (pwMS) or healthy controls (HC), and 40 proteins were expressed above the LOD in more than 75% of the samples. While it is true that some proteins (GZMB, CLEC6A, SETMAR, CRTAM) exhibit less than 50% of samples above the LOD, these proteins have not been selected in any of our final models and therefore are not suggested as predictive biomarkers in our study. The top performing proteins for diagnosis (MZB1, CD79B, CD27, TNFRSF13B, IL-12p40) and predicting disease activity over two years (NfL, IL-1RA, CCL3), consistently exhibit more than 75% (in fact more than 95%) of samples above the LOD for both pwMS and HC samples.

In the model for predicting disability worsening, the majority of the included proteins (8 out of 11) were measured above the LOD in at least 75% of samples from pwMS. The remaining

three proteins (SLAMF7, TYMP, FYB1) in this model had at least 60% of the samples above the LOD for pwMS. To be precise, the percentage of samples from pwMS in the discovery cohort above LOD are: NfL (100%), ICAM3 (100%), LTA (100%), CHI3L1 (100%), TNFRSF1B (100%), LY9 (90%), CXCL13 (84%), FCN2 (79%), FYB1 (70%), TYMP (66%), SLAMF7 (61%). This demonstrates that the proteins in our final models are consistently detected in high proportions of our samples. We agree that the three proteins (SLAMF7, TYMP, FYB1) may not be suitable as standalone biomarkers due to their lower detection rate in specific groups, as the reviewer has rightly pointed out, but they hold potential value in combination with other markers especially since they are detected above the LOD in at least 60% of the samples. To further investigate the value of these three markers for the disability worsening model, we assessed the performance of models with different combinations of the three proteins removed. If all three proteins were removed the Spearman's correlation coefficient (SCC) was 0.61 in the discovery cohort and 0.54 in the replication cohort. If two of these proteins were removed the SCC was in the range [0.62, 0.65] in the discovery cohort and [0.55, 0.62] in the replication cohort, and if one of these proteins were removed the SCC was in the range [0.65, 0.68] in the discovery cohort and [0.59, 0.72] in the replication cohort.

To ensure complete transparency regarding this issue, we have incorporated the detection rates into our manuscript (line 182, 230, and 329-331 in the results section) and added the performance results of our disability worsening model after the exclusion of these three proteins in Supplementary Table 5.

Reviewer #3 (expert in neurology and biomarkers for neurological diseases - this reviewer was asked to review the authors' responses to Reviewer #1):

The authors have responded dilligently and convincingly to the reviewer's comments and have adapted their manuscript where necessary. This has further improved the quality of the manuscript that is certainly documenting in a balanced way an important step forward in the difficult field of proteomics as a tool for classification and prognostication of MS.

Response:

We are pleased to hear that our responses to the previous comments have been convincing and have contributed to improving the quality of our manuscript. We appreciate the reviewer's positive feedback and thoughtful evaluation. We express our gratitude to the reviewer for recognizing the importance of identifying protein biomarkers for classification and prognostication of MS.

REVIEWERS' COMMENTS

Reviewer #2 (Remarks to the Author):

The authors have provided additional information regarding the translatability of some of the identified biomarkers suffering a high percentage of measurements below the LOD. This addressed the last part of my questions and suggestions. I would like to congratulate the authors on their work and I am looking forward to learning about follow-up validation studies.

RESPONSE TO REVIEWERS' COMMENTS

Reviewer #2 (Remarks to the Author):

The authors have provided additional information regarding the translatability of some of the identified biomarkers suffering a high percentage of measurements below the LOD. This addressed the last part of my questions and suggestions. I would like to congratulate the authors on their work and I am looking forward to learning about follow-up validation studies.

Response:

We sincerely appreciate the reviewer's constructive feedback on our manuscript and are happy that the additional information we provided concerning the translatability of certain biomarkers, has addressed their concerns. Indeed, we share the reviewer's anticipation of future validation studies. Once again, we thank the reviewer for the valuable feedback which has helped us improve our manuscript and for taking the time to review our work.